# An intestinally secreted host factor promotes microsporidia invasion of *C. elegans*

**Hala Tamim El Jarkass[1], Calvin Mok[1], Michael R Schertzberg[2], Andrew G Fraser[1], Emily R Troemel[3], Aaron W Reinke[1]\***

[1]Department of Molecular Genetics, University of Toronto, Toronto, Canada; [2]The Donnelly Centre, University of Toronto, Toronto, Canada; [3]Division of Biological Sciences, University of California, San Diego, La Jolla, United States

**Abstract** Microsporidia are ubiquitous obligate intracellular pathogens of animals. These parasites often infect hosts through an oral route, but little is known about the function of host intestinal proteins that facilitate microsporidia invasion. To identify such factors necessary for infection by *Nematocida parisii*, a natural microsporidian pathogen of *Caenorhabditis elegans*, we performed a forward genetic screen to identify mutant animals that have a Fitness Advantage with *Nematocida* (Fawn). We isolated four *fawn* mutants that are resistant to *Nematocida* infection and contain mutations in *T14E8.4*, which we renamed *aaim-1* (Antibacterial and Aids invasion by Microsporidia). Expression of AAIM-1 in the intestine of *aaim-1* animals restores *N. parisii* infectivity and this rescue of infectivity is dependent upon AAIM-1 secretion. *N. parisii* spores in *aaim-1* animals are improperly oriented in the intestinal lumen, leading to reduced levels of parasite invasion. Conversely, *aaim-1* mutants display both increased colonization and susceptibility to the bacterial pathogen *Pseudomonas aeruginosa* and overexpression of *aaim-1* reduces *P. aeruginosa* colonization. Competitive fitness assays show that *aaim-1* mutants are favored in the presence of *N. parisii* but disadvantaged on *P. aeruginosa* compared to wild-type animals. Together, this work demonstrates how microsporidia exploits a secreted protein to promote host invasion. Our results also suggest evolutionary trade-offs may exist to optimizing host defense against multiple classes of pathogens.

**\*For correspondence:**
aaron.reinke@utoronto.ca

**Competing interest:** The authors declare that no competing interests exist.

## Editor's evaluation

Host factors involved in microsporidia infection have not been characterized in any detail, despite their ubiquitous distribution as infectious agents of animals. Through genetics screens, this study identifies a gene required for microsporidia invasion of the *C. elegans* intestine. Decreased susceptibility to microsporidia comes at the cost of increased colonization and infection by certain bacterial species, suggesting a tradeoff between resistance to different classes of pathogens.

## Introduction

Microsporidia are a large group of obligate intracellular parasites that infect most types of animals (*Murareanu et al., 2021b*). These ubiquitous parasites possess the smallest known eukaryotic genomes and are extremely reliant on their host as a result of the loss of many genes involved in metabolism and energy production (*Corradi, 2015*; *Wadi and Reinke, 2020*). Microsporidia can have a large impact on the evolution of their hosts, as infection often leads to a reduction in host offspring and the effect of this selective pressure has resulted in resistant animals within a population (*Balla et al., 2015*; *Routtu and Ebert, 2015*). Microsporidia are currently a major threat to many commercially

important species such as honeybees and shrimp (*Martín-Hernández et al., 2018*; *Jaroenlak et al., 2018*). Many species also infect humans and infections in immunocompromised individuals can result in lethality (*Stentiford et al., 2016*). Despite their ubiquitous nature, effective treatment strategies are currently lacking for these poorly understood parasites (*Han and Weiss, 2018*).

Microsporidia infection begins with invasion of host cells. Microsporidia possess fascinating invasion machinery, a unique structure known as the polar tube (*Han et al., 2020*). This apparatus, resembling a long thread, is often coiled within a dormant spore. However, once inside of a host, and in proximity to the tissue of interest, the polar tube rapidly emerges or 'fires', releasing the infectious material (the sporoplasm) which is deposited intracellularly either through direct injection, or through the internalization of the sporoplasm (*Han et al., 2020*; *Tamim El Jarkass and Reinke, 2020*).

A number of microsporidia proteins have been demonstrated to play important roles during invasion by insect- and human-infecting species of microsporidia (*Han et al., 2020*). For example, spore wall proteins can interact with host cells through the recognition of sulfated glycosaminoglycans, heparin binding motifs, integrins, and proteins on the cell surface (*Hayman et al., 2001*; *Hayman et al., 2005*; *Southern et al., 2007*; *Li et al., 2009*; *Wu et al., 2008*; *Chen et al., 2017*). In *Encephalitozoon* species, polar tube proteins (PTP) can mediate interactions with the host. For instance, O-linked mannosylation on PTP1 has been demonstrated to bind mannose binding receptors, whereas PTP4 interacts with the transferrin receptor (Trf1) (*Tamim El Jarkass and Reinke, 2020*; *Xu et al., 2003*; *Xu et al., 2004*; *Han et al., 2017*). Additionally, the sporoplasm surface protein, EhSSP1, binds to an unknown receptor on the cell surface (*Han et al., 2019*). These proteins on the spore, polar tube, and sporoplasm have all been shown to promote microsporidia adhesion or invasion of host cells in culture systems, but the role of these proteins during animal infection is unclear.

The nematode *Caenorhabditis elegans* is infected in its natural habitat by several species of microsporidia, and frequently by *Nematocida parisii* (*Luallen et al., 2016*; *Zhang et al., 2016*; *Troemel et al., 2008*). This species infects the intestinal cells of *C. elegans,* which possess similarity to those of mammalian cells, making this animal both a relevant tissue and model to study these infections in vivo (*Troemel et al., 2008*; *Troemel, 2011*). Infection of *C. elegans* by *N. parisii* begins when spores are consumed by the worm, where they then pass through the pharynx into the intestinal lumen and fire, depositing sporoplasms inside of intestinal cells. Within 72 hr, the sporoplasm will divide into meronts, which differentiate into spores, that then exit the animal, completing the parasite's life cycle (*Balla et al., 2016*; *Willis et al., 2021*). Infection with *N. parisii* leads to reduced fecundity and premature mortality in *C. elegans* (*Troemel et al., 2008*; *Balla et al., 2016*). Several mutants have been shown to affect proliferation and spore exit (*Botts et al., 2016*; *Szumowski et al., 2014*). Immunity that can either prevent infection or clear the pathogen once infected has also been described (*Balla et al., 2015*; *Willis et al., 2021*; *Balla et al., 2019*; *Reddy et al., 2019*; *Tecle et al., 2021*). In contrast, very little is known about how *N. parisii* invades *C. elegans* intestinal cells. Almost all the microsporidia proteins known to facilitate invasion of host cells are not conserved in *N. parisii* and although host invasion factors described in other species are present in *C. elegans*, there is no evidence that they are being used by microsporidia during invasion of *C. elegans* (*Tamim El Jarkass and Reinke, 2020*).

To understand how microsporidia invade animal cells, we performed a forward genetic screen to identify host factors that promote infection. We identified a novel, nematode-specific protein, AAIM-1, whose loss of function confers resistance to *N. parisii* infection. This protein is expressed in intestinal cells, secreted into the intestinal lumen, and is necessary to ensure proper spore orientation during intestinal cell invasion. In addition, we show that AAIM-1 limits bacterial colonization of pathogenic *Pseudomonas aeruginosa*. Strikingly, AAIM-1 plays opposing roles on host fitness in the face of pathogenesis. The utilization of a host factor critical for bacterial defense reflects a clever strategy to ensuring microsporidia's reproductive success.

## Results

### A forward genetic screen identifies *aaim-1* as being necessary for *N. parisii* infection

To identify host factors needed for infection by microsporidia, we carried out a forward genetic screen using a *C. elegans* model of *N. parisii* infection. We took advantage of the previously described phenotypes of *C. elegans* displaying reduced fitness when infected with *N. parisii,* including reduced

progeny production and stunted development (*Balla et al., 2016*; *Willis et al., 2021*; *Luallen et al., 2015*; *Murareanu et al., 2021a*). We mutagenized animals and subjected their F2 progeny to *N. parisii* infection. After infecting populations for five subsequent generations, we selected individual worms containing embryos, indicating increased fitness in the presence of infection (see Materials and methods). We identified four independent isolates that when exposed to *N. parisii* reproducibly had higher fractions of animals containing embryos compared to wild type (N2). We named these isolates Fitness Advantage With *Nematocida* (*fawn* 1–4) (*Figure 1—figure supplement 1a*).

As *C. elegans* that are less infected with *N. parisii* produce more progeny, we hypothesised that these *fawn* mutants would be resistant to *N. parisii* infection (*Willis et al., 2021*). To determine this, we grew the three isolates with the strongest phenotype, *fawn* 1–3, in the presence and absence of *N. parisii*, and stained each population of worms with the chitin binding dye, Direct-yellow 96 (DY96), at 72 hr post infection (hpi). DY96 allows for the visualization of chitinous microsporidia spores as well as worm embryos (*Figure 1a*). In the absence of infection, there is no difference in the fraction of *fawn-2* and *fawn-3* animals developing into adults containing embryos (gravid adults), although *fawn-1* has a modest defect. In comparison, all three *fawn* isolates generate significantly more gravid adults than N2 animals in the presence of infection (*Figure 1b*). We next examined the fraction of animals in each strain containing intracellular microsporidia spores and observed that all three *fawn* isolates display significantly fewer numbers of spore-containing worms (*Figure 1c*). These results suggest that *fawn* mutants are missing an important factor for efficient microsporidia infection.

To identify the causal mutations underlying the Fawn phenotype, we used a combination of whole-genome sequencing and genetic mapping. We generated F2 recombinants and performed two rounds of infection with microsporidia, selecting for gravid animals. After each round, we used molecular inversion probes to determine the region of the genome linked to the causal mutation (*Mok et al., 2017*). This revealed strong signatures of selection on the left arm of chromosome X in all three *fawn* isolates and absent in N2 (*Figure 1—figure supplement 1b*). Analysis of whole genome sequencing showed that all four *fawn* isolates contained different alleles of *T14E8.4*, which we named *aaim-1* (Antibacterial and Aids Invasion by Microsporidia-1) for reasons described below (*Figure 1d*). We validated the role of *aaim-1* in resistance to infection using several additional alleles: an independent allele *aaim-1* (*ok295*), carrying a large gene deletion in both *aaim-1* and *dop-3*, and a CRISPR-Cas9 derived allele, *aaim-1* (*kea22*), that contains a large gene deletion. Both of these alleles displayed a fitness advantage when infected with *N. parisii* (*Figure 1d and e* and *Figure 1—figure supplement 1c, d*). These data demonstrate that *aaim-1* is the causative gene underlying the *fawn* 1–4 infection phenotypes. In subsequent experiments we utilized both *aaim-1 (kea22)*, and *fawn-3 (kea28),* carrying a 2.2 kb deletion in *aaim-1,* which was outcrossed to N2 six times [hereafter referred to as *aaim-1 (kea28)*].

## *aaim-1* is expressed in the pharynx and intestine, and secretion is important for function

AAIM-1 is a poorly characterized protein that does not possess any known or conserved domains. Homologs of the protein exist in both free-living and parasitic nematodes (*Figure 2—figure supplement 1*). To further characterize the role of AAIM-1 during *N. parisii* infection, we generated transgenic extrachromosomal lines of *C. elegans* carrying a reporter transgene of GFP under control of the *aaim-1* promoter. GFP fluorescence was observed in the terminal bulb of the pharynx as well as the posterior of the intestine throughout development (*Figure 2a*). Embryos and L1 animals display additional expression in the arcade cells of the pharynx (*Figure 2a* and *Figure 2—figure supplement 2a*).

The first 17 amino acids of AAIM-1 are predicted to encode a signal peptide (*Almagro Armenteros et al., 2019*). This suggests that AAIM-1 may be secreted into the pharyngeal and intestinal lumen, the extracellular space through which *N. parisii* spores pass before invading intestinal cells. To test which tissues AAIM-1 functions in and if secretion is important for function, we generated a series of transgenic worms expressing extrachromosomal arrays (Key resources table, Appendix 1). First, we generated transgenic *aaim-1 (kea22)* animals expressing AAIM-1 tagged on the C-terminus with a 3 x Flag epitope. Transgenic animals expressing AAIM-1 under its native promoter complement the ability of *aaim-1 (kea22)* animals to develop into adults in the presence of a high amount of *N. parisii* spores (*Figure 2b*). A construct expressing GFP or GFP::3xFlag does not influence this phenotype nor does the presence of the epitope tag impair the ability of AAIM-1 to rescue the mutant

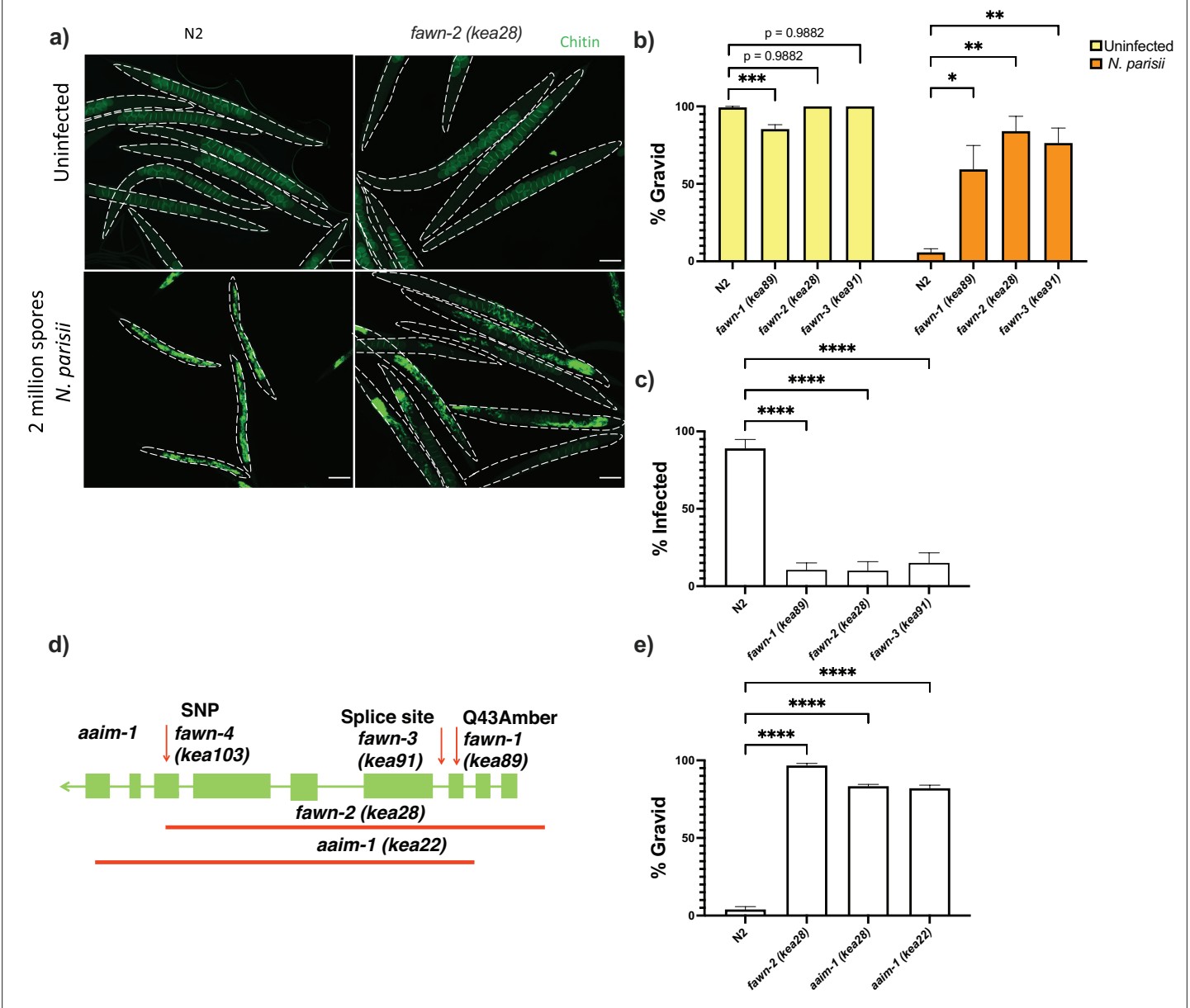

**Figure 1.** Mutations in *aaim-1* result in resistance to *N. parisii* infection. (**a-c**, and **e**) L1 stage wild-type (N2) and *aaim-1* mutant animals were infected with either a high dose (a, b, and e) or a very lose dose (**c**) of *N. parisii*, fixed at 72 hr, and stained with direct-yellow 96 (DY96). (**a**) Representative images stained with DY96, which stains *C. elegans* embryos and microsporidia spores. Scale bars, 100 µm. (**b and e**) Graph displays percentage of gravid worms. (**c**) Percentage of worms that contain newly formed *N. parisii* spores. (**d**) Schematic depicting the nature and location of the different *aaim-1* alleles. Boxes represent exons, and connecting lines represent introns. Arrows depict point mutations, and the solid red lines depict deletions. *fawn-2 (kea28)* has a 2.2 kb deletion and *aaim-1 (kea22)* has a 2.3 kb deletion. *fawn-1 (kea89)* carries a C127T, Q43Stop mutation, *fawn-3 (kea91)* carries a G221A splice site mutation and *fawn-4 (kea103)* carries a C1286T, A429V mutation in *aaim-1*. (b,c, and e) Data is from three independent replicates of at least 90 worms each. Mean ± SEM represented by horizontal bars. p-Values determined via one-way ANOVA with post hoc. Significance defined as: * p < 0.05, ** p < 0.01, *** p < 0.001, **** p < 0.0001.

The online version of this article includes the following source data and figure supplement(s) for figure 1:

**Source data 1.** Mutations in *aaim-1* result in resistance to *N. parisii* infection.

**Figure supplement 1.** Mapping and validation of *aaim-1* as the gene associated with resistance to *N. parisii*.

**Figure supplement 1—source data 1.** Mapping and validation of *aaim-1* as the gene associated with resistance to *N. parisii*.

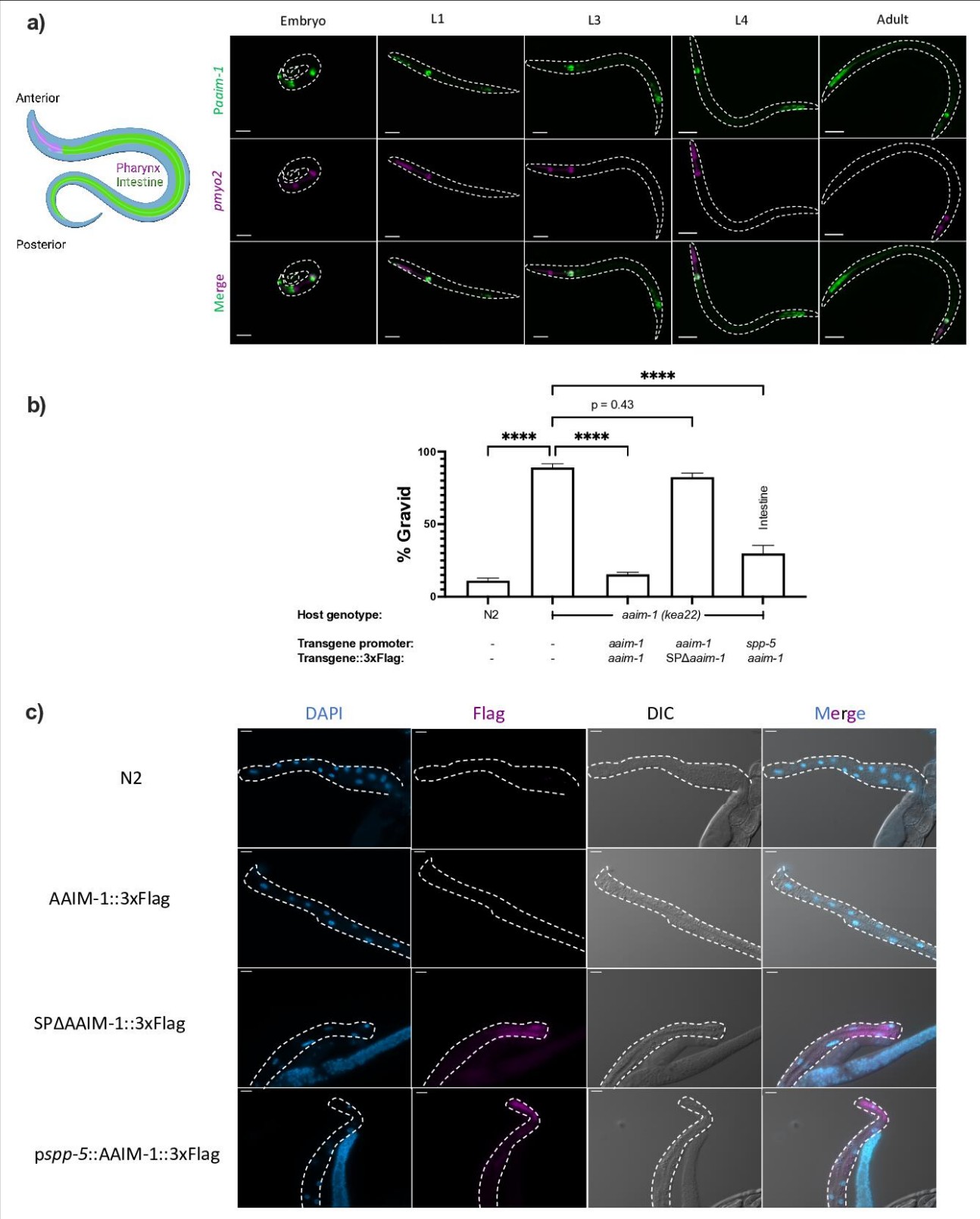

**Figure 2.** AAIM-1 is secreted from intestinal cells. (**a**) Wild-type worms containing an extrachromosomal array expressing GFP from the *aaim-1* promoter and mCherry (labelled in magenta) in the pharyngeal muscles were imaged at the embryo, L1, L3, L4, and adult stage. Embryo, L1, and L3 animals were imaged at 40 x, scale bar 20 μm and L4 and adult animals were imaged at 20 x, scale bar 50 μm. L1 to L4 animals are oriented anterior to posterior and the adult animal is oriented posterior to anterior from left to right. Schematic made with Biorender.com (**b**) N2, *aaim-1, and aaim-1* expressing

*Figure 2 continued on next page*

*Figure 2 continued*

extrachromosomal arrays were infected with a medium-2 dose of *N. parisii*. Graph displays percentage of gravid worms. Data is from three independent replicates of at least 90 worms each. Mean ± SEM represented by horizontal bars. p-Values determined via one-way ANOVA with post hoc. Significance defined as **** p < 0.0001 (**c**) Intestines (denoted by dashed lines) of 72 hr post-L1 adults were dissected and stained using anti-Flag (magenta) and DAPI (blue). Images taken at 40 x, scale bar 20 μm.

The online version of this article includes the following source data and figure supplement(s) for figure 2:

**Source data 1.** AAIM-1 is secreted from intestinal cells.

**Figure supplement 1.** AAIM-1 is conserved in both free-living and parasitic nematodes.

**Figure supplement 2.** *aaim-1* is expressed in arcade cells and presence of C-terminal 3 x Flag tag does not disrupt AAIM-1 function.

**Figure supplement 2—source data 1.** *aaim-1* is expressed in arcade cells and presence of C-terminal 3 x Flag tag does not disrupt AAIM-1 function.

phenotype (*Figure 2—figure supplement 2b*). We next generated a signal peptide mutant allele of AAIM-1 missing the first 17 amino acids (*SPΔaaim-1*), which is unable to complement the *aaim-1 N. parisii* infection phenotype. In contrast, AAIM-1 expressed from an intestinal-specific promoter (*spp-5*) (*Reinke et al., 2017*) can rescue the infection phenotype of *aaim-1* (*kea22*) (*Figure 2b*).

To determine where AAIM-1 localizes, we dissected the intestines from transgenic worms and performed immunofluorescence using anti-Flag antibodies. We were unable to detect expression of AAIM-1::3xFlag when expressed from its endogenous promoter. However, we observed protein expression in the intestinal cells of animals expressing AAIM-1::3xFlag from a strong, intestinal specific-promoter or when the signal peptide was removed (*Figure 2c*). We did not observe AAIM-1::3xFlag localized in the extracellular space of the intestinal lumen, possibly due to rapid turnover of intestinal contents or due to loss from dissection of the intestines (*McGhee and Ghafouri, 2007*). The increased expression in the signal peptide mutant suggests an accumulation of protein that is unable to be secreted. Taken together, these data demonstrate that AAIM-1 is secreted and acts within the intestinal lumen to promote *N. parisii* infection.

## AAIM-1 is only necessary for microsporidia infection at the earliest larval stage

*N. parisii* infection of *C. elegans* can occur throughout development, but several forms of immunity toward microsporidia have been shown to be developmentally regulated (*Balla et al., 2015*; *Willis et al., 2021*). To determine if *aaim-1* mutant animals display developmentally restricted resistance to infection, we infected *fawn* 1–3 at the L1 and L3 stage. For these experiments, we took advantage of another intestinal-infecting species of microsporidia, *Nematocida ausubeli*, which has a more severe effect on *C. elegans* fecundity, allowing us to determine fitness defects after the L1 stage (*Balla et al., 2015*; *Zhang et al., 2016*; *Balla et al., 2016*). *fawn* isolates are resistant to *N. ausubeli* as seen by an increase in the fraction of gravid adults in the population after exposure to a medium dose of *N. ausubeli* (*Figure 3a*). When we initiated infections at the L3 stage of growth, *fawn* isolates do not have increased resistance, and instead exhibit wild-type levels of susceptibility (*Figure 3b*). To rule out the possibility that this L1 restricted phenotype was the result of exposure to sodium hypochlorite treatment, which we used to synchronize worms, we exposed embryos that were naturally laid by adults within a 2-hr window to *N. parisii* infection. Animals synchronized in this manner still display a robust resistance to *N. parisii* (*Figure 3—figure supplement 1c*). Thus, resistance to infection in *aaim-1* mutants is developmentally restricted and AAIM-1 is utilized by several species of microsporidia.

## AAIM-1 is needed for efficient invasion of intestinal cells

Resistance to infection could be the result of a block in invasion, proliferation, or through the destruction of the parasite. To test the mechanism of resistance in *aaim-1* mutants, we performed pulse-chase infection assays at the L1 and L3 stage of development (*Balla et al., 2015*; *Willis et al., 2021*). Here, we treated animals with a medium-1 dose (as defined in *Supplementary file 1*) of *N. parisii* for 3 hr, washed away any un-ingested spores, and then replated the animals in the absence of spores for an additional 21 hr. We then used an 18 S RNA Fluorescent In Situ Hybridization (FISH) probe to detect *N. parisii* sporoplasms, which is the earliest stage of microsporidia invasion. In our *fawn* 1–3 isolates, we detect less invasion at 3 hpi compared to N2 (*Figure 3—figure supplement 1a*). However, there was no reduction in the number of infected animals between 3 hpi and 21

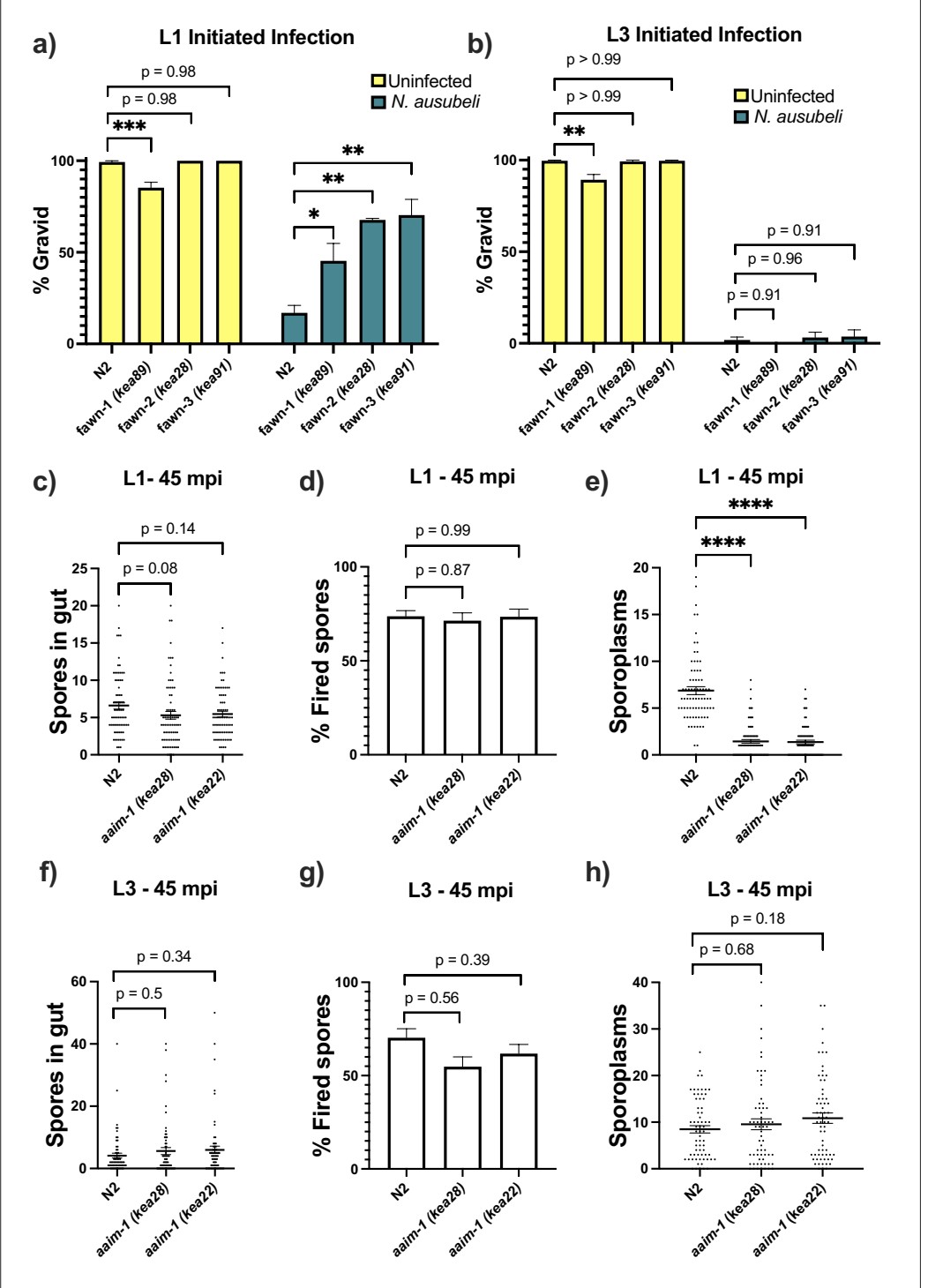

**Figure 3.** *aaim-1* mutants are resistant to microsporidia at the earliest larval stage due to spore misfiring. (**a–b**) N2 and *aaim-1* mutants were infected with a medium dose of *N. ausubeli* at either the L1 stage for 72 hr (**a**) or a high dose of *N. ausubeli* at the L3 stage for 48 hr (**b**) Graph displays percentage of gravid worms. (**c–f**) N2 and *aaim-1* animals were infected with a medium-3 dose of *N. parisii* for 45 min at L1 (**c–e**) or L3 (**f–h**), fixed, and then stained with DY96 and an *N. parisii* 18 S RNA fish probe. The number of spores per animal (**c,f**), the percentage of spores fired (**d,g**), and the number of sporoplasm per worm (**e,h**) are displayed. Data is from three independent replicates of at least 100 worms each (**a–b**) or 20–30 worms each (**c–h**). (**a–b**) Mean ± SEM represented by horizontal bars. p-values determined via one-way ANOVA with post hoc. Significance defined as: * p < 0.05, ** p < 0.01, *** p < 0.001, **** p < 0.0001.

*Figure 3 continued on next page*

*Figure 3 continued*

The online version of this article includes the following source data and figure supplement(s) for figure 3:

**Source data 1.** *aaim-1* mutants are resistant to microsporidia at the earliest larval stage due to spore misfiring.

**Figure supplement 1.** *aaim-1* mutants do not clear *N. parisii* and developmentally restricted *N. parisii* invasion defect is not due to a feeding defect.

**Figure supplement 1—source data 1.** *aaim-1* mutants do not clear *N. parisii* and developmentally restricted *N. parisii* invasion defect is not due to a feeding defect.

**Figure supplement 2.** Invasion defects in *aaim-1* only occur at the L1 stage of development and a mutation in *aaim-1* does not alter the width of the intestinal lumen.

**Figure supplement 2—source data 1.** Invasion defects in *aaim-1* only occur at the L1 stage of development and a mutation in *aaim-1* does not alter the width of the intestinal lumen.

hpi, indicating that pathogen clearance was not occurring. This defect in invasion was not present at the L3 stage, providing further support that resistance is restricted to the L1 stage in *aaim-1* mutants (*Figure 3—figure supplement 1b*). A reduction in invasion could be due to a feeding defect, leading to a reduction in spore consumption. To test rates of consumption, we measured the intestinal accumulation of fluorescent beads. We find that *aaim-1* alleles displayed wild-type levels of bead accumulation, unlike the feeding defective strain *eat-2 (ad465)* (*Figure 3—figure supplement 1d*).

For *N. parisii* to invade host cells, spores must first enter the intestinal lumen and fire their polar tube (*Willis et al., 2021*). To test if *aaim-1* mutants have defects in spore entry or spore firing, we infected animals for either 45 min or 3 hr, at the L1 and L3 stages. We then fixed and stained animals with both an *N. parisii* 18 S RNA FISH probe and DY96 and quantified the number of spores present in the intestinal lumen of animals. Here, *aaim-1* animals infected for 45 min or 3 hr at L1 or L3 contained similar amounts of spores as N2 animals (*Figure 3c and f*, and *Figure 3—figure supplement 2a,d*). The percentage of fired spores present within these animals is also not significantly different at either developmental stage (*Figure 3d and g* and *Figure 3—figure supplement 2b,e*). We then counted the number of sporoplasms per animal and observed significantly fewer invasion events in *aaim-1* mutant animals infected at L1 (*Figure 3e* and *Figure 3—figure supplement 2c*). In contrast, the number of sporoplasms in L3 stage *aaim-1* alleles are similar to that observed in the N2 strain (*Figure 3h* and *Figure 3—figure supplement 2f*). These results demonstrate that the *N. parisii* invasion defect in *aaim-1* mutants is not caused by differences in spore firing or accumulation. Instead, these results suggest that spores are misfiring, leading to unsuccessful parasite invasion.

## AAIM-1 plays a role in promoting proper spore orientation

To determine how AAIM-1 promotes *N. parisii* invasion, we further examined the invasion process. We pre-stained spores with calcofluor white (CFW) and assessed their orientation relative to the intestinal apical membrane using the apical membrane marker PGP-1::GFP in L1 worms infected for 45 min (*Figure 4a*). In N2 animals, 32.4% of spores are angled relative to the apical membrane. In contrast, spores in an *aaim-1* mutant were angled 14.3% of the time (*Figure 4b*). Several host factors that promote microsporidia invasion cause adherence to host cells (*Tamim El Jarkass and Reinke, 2020*). To determine if AAIM-1 influences the location of spores relative to intestinal cells in *aaim-1* mutants, we measured the perpendicular distance from the center of a parallel spore to the apical membrane of the intestine. Surprisingly, parallel spores in *aaim-1* alleles were significantly closer to the apical membrane (0.29 µm) than those in N2 (0.34 µm) (*Figure 4c*). In agreement with resistance being developmentally restricted, *aaim-1* mutants display wild-type spore orientations and distances from the membrane when infections were initiated at the L3 stage (*Figure 4d and e*). The width of the intestinal lumen at the L1 stage does not differ significantly between N2 and *aaim-1* mutants, however, L3 animals generally possess wider intestinal lumens (*Figure 3—figure supplement 2g,h*). Thus, taken together these results suggest that AAIM-1 plays a distinct role in the intestinal lumen at L1 to promote proper spore orientation, through maintaining an appropriate distance and angle to the apical membrane, resulting in successful invasion.

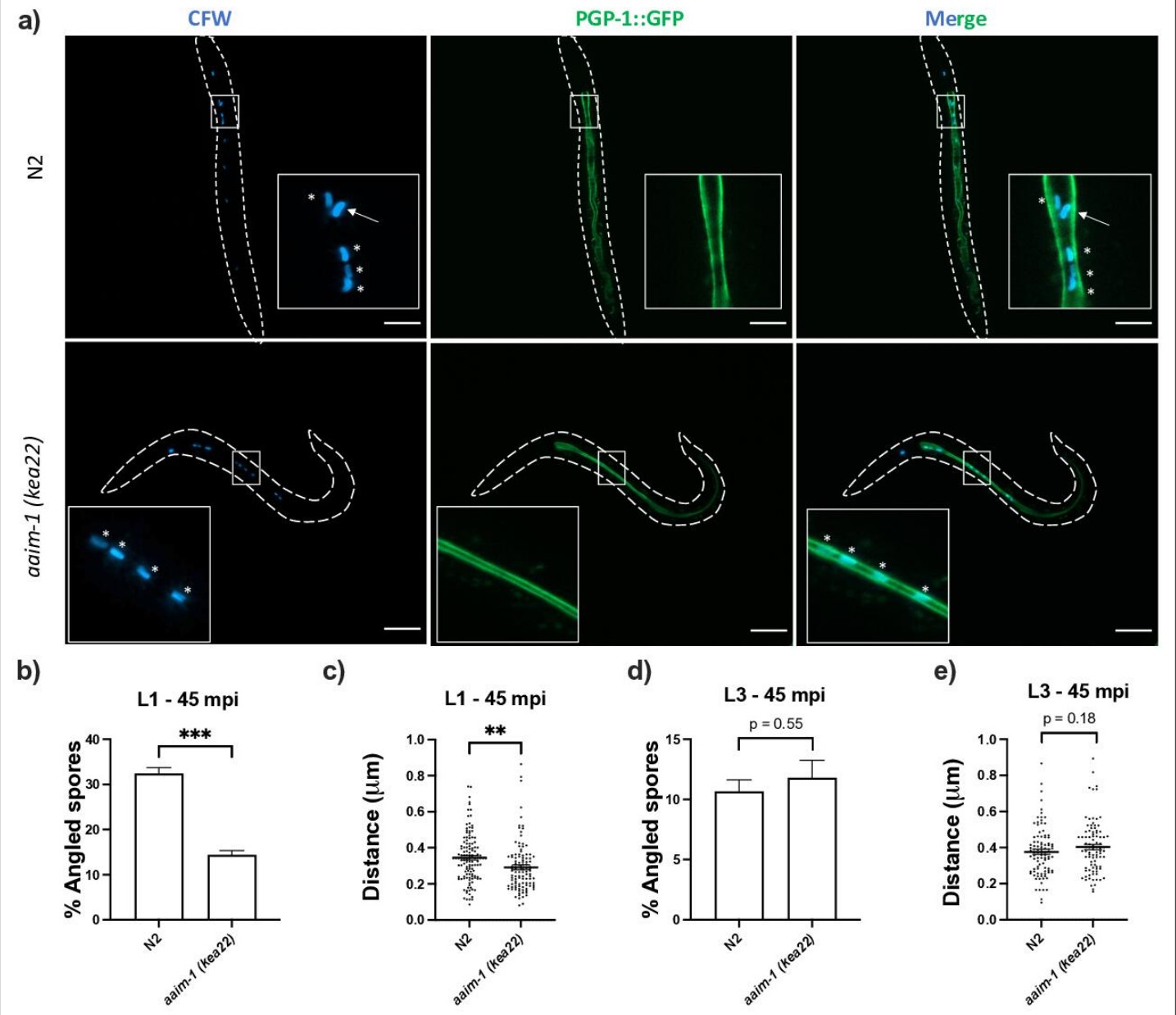

**Figure 4.** Spores in *aaim-1* mutants display improper orientation and distance to the apical intestinal membrane. (**a–e**) PGP-1::GFP and *aaim-1(kea22)*; PGP-1::GFP animals were infected with a very high dose of calcofluor white (CFW) pre-stained *N. parisii* spores for 45 min at either the L1 stage (**a–c**) or the L3 stage (**d-e**). (**a**) Representative images of live animals containing stained spores (blue) relative to the apical intestinal membrane (GFP). Arrow indicates an example of an angled spore and asterisks indicate parallel spores. Images taken at 63 x, scale bar 20 μm. (**b, d**) Percentage of angled spores. Data is from three independent replicates of at least 90 spores each. (**c, e**) Distance from the center of each spore to the intestinal apical membrane. Data is from three independent replicates of at least 25 spores each. Mean ± SEM represented by horizontal bars. p-Values determined via unpaired Student's t-test. Significance defined as ** p < 0.01, *** p < 0.001.

The online version of this article includes the following source data for figure 4:

**Source data 1.** Spores in *aaim-1* mutants display improper orientation and distance to the apical intestinal membrane.

## AAIM-1 inhibits intestinal colonization by *Pseudomonas aeruginosa*

Interestingly, *aaim-1* has been shown to be upregulated by a variety of different fungal and bacterial pathogens, including *P. aeruginosa*. (*Engelmann et al., 2011*; *Head et al., 2017*) Using our transcriptional reporter strain, we sought to confirm this and determine if microsporidia infection could also induce *aaim-1* transcription. N2 animals carrying a transcriptional reporter (p*aaim-1*::GFP::3xFlag)

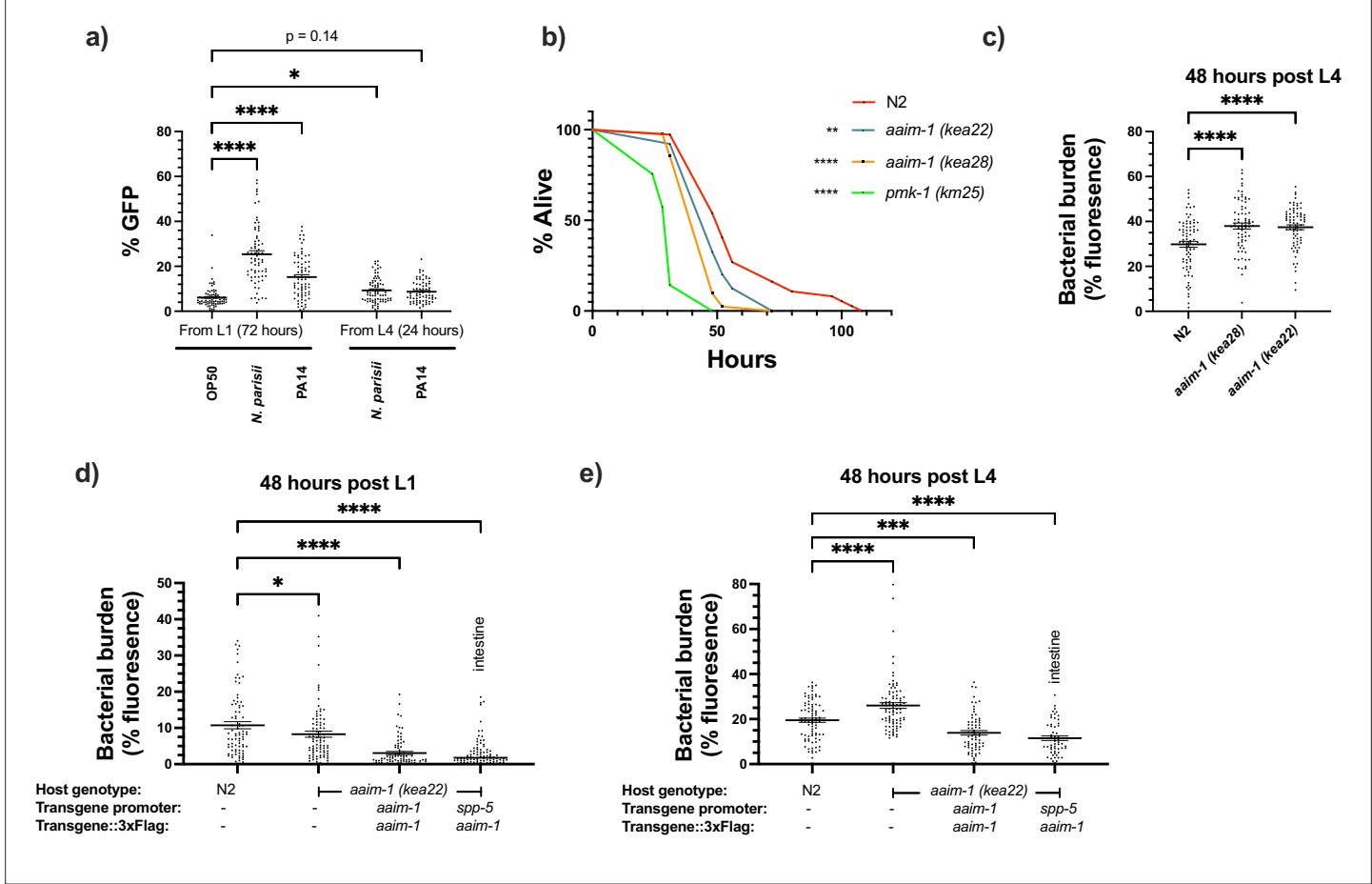

**Figure 5.** *aaim-1* is upregulated by *N. parisii and P. auerginosa* and *aaim-1* animals are susceptible to infection by *P. aeruginosa*. (**a**) Expression of p*aaim-1*::GFP::3xFlag in response to infection with either PA14 or *N. parisii* for either 72 hr from L1 or 24 hr from L4. Data is from three independent replicates of at least 18–25 worms each. Every point represents a single worm. Percentage GFP was measured as the percentage of the animal containing GFP via FIJI. (**b**) L4 stage N2, *aaim-1*, and *pmk-1 (km25)* animals were plated on full lawns of *P. aeruginosa* PA14::DsRed and the percentage of animals alive was counted over the course of 96 hr. $TD_{50}$: N2 48 hr, aaim-1 *(kea28)* 44 hr, *aaim-1 (kea22)* 33 hr, and *pmk-1 (km25)* 28 hr. Three independent replicates were carried out, and a representative replicate is displayed. At least 37 worms were quantified per strain. p-Values determined via Log-rank (Mantel-Cox) test. Significance defined as * $p < 0.05$, ** $p < 0.01$. (**c–e**) N2, *aaim-1*, or *aaim-1* animals with different extrachromosomal arrays were plated on PA14::DsRed at either the L1 stage (**d**) or L4 stage (**c,e**) for 48 hr. Bacterial burden was measured as the percentage of the animal containing PA14::DsRed. Data is from three independent replicates of 20–30 worms each. Every point represents a single worm. Mean ± SEM represented by horizontal bars. p-Values determined via two-way (**a**) or one-way ANOVA(c-e) with post hoc. Significance defined as * $p < 0.05$, ** $p < 0.01$, *** $p < 0.001$, **** $p < 0.0001$.

The online version of this article includes the following source data and figure supplement(s) for figure 5:

**Source data 1.** *aaim-1* is upregulated by *N. parisii and P. auerginosa* and *aaim-1* animals are susceptible to infection by *P. aeruginosa*.

**Figure supplement 1.** Susceptibility to *P. aeruginosa* PA14 appears at L4.

**Figure supplement 1—source data 1.** Susceptibility to *P. aeruginosa* PA14 appears at L4.

**Figure supplement 2.** A mutation in *aaim-1* does not influence *C. elegans* defense against *S. aureus* or lifespan.

**Figure supplement 2—source data 1.** A mutation in *aaim-1* does not influence *C. elegans* defense against *S. aureus* or lifespan.

were exposed to *N. parisii*, *P. aeruginosa* PA14, or *E. coli* OP50-1, and the levels of GFP quantified when grown on these pathogens for 72 hr from the L1 stage, or for 24 hr from the L4 stage. Infection by either *N. parisii* or *P. aeruginosa* PA14 resulted in the upregulation of *aaim-1* as detected by an increase in the GFP signal (***Figure 5a*** and ***Figure 5—figure supplement 1f***).

Previously, an *aaim-1* deletion strain, RB563 *(ok295)*, was shown to display reduced survival on lawns of *P. aeruginosa* PA14 (***Styer et al., 2008***). The enhanced susceptibility previously reported was attributed to *dop-3*, which is also partially deleted in RB563 *(ok295)* (***Styer et al., 2008***). To determine

if *aaim-1* mutants are susceptible to pathogenic bacterial infection, we assayed the survival of L4 stage worms in *P. aeruginosa* PA14 slow killing assays. A mutant in the p38 MAPK pathway (*pmk-1*) was used as control for susceptibility to PA14 (*Kim et al., 2002*). We observed reduced survival in *aaim-1* alleles, although not to the same extent as the *pmk-1* mutant (*Figure 5b* and *Figure 5—figure supplement 1a-c*). In contrast to significant susceptibility to the Gram-negative *P. aeruginosa*, *aaim-1* mutants do not display enhanced susceptibility to the Gram-positive bacterium *Staphylococcus aureus* NCTC8325, suggesting specificity of AAIM-1 to PA14 infection (*Figure 5—figure supplement 2a*).

Lethality in slow killing assays is a result of *P. aeruginosa* accumulation within the intestinal lumen (*Tan et al., 1999*; *Kirienko et al., 2014*). To investigate if *aaim-1* alleles displayed higher levels of bacterial burden, animals were grown on lawns of PA14::DsRed at the L1 or L4 stage for 48 hr. *aaim-1* mutant alleles exposed as L4s, but not L1s, displayed higher bacterial burden relative to N2 (*Figure 5c* and *Figure 5—figure supplement 1d,e*). To test if intestinal expression of *aaim-1* was sufficient to limit bacterial colonization, transgenic *aaim-1 (kea22)* overexpressing AAIM-1::3xFlag from the endogenous or an intestinal-specific promoter were exposed to lawns of PA14::DsRed. When grown for 48 hr at the L1 or L4 stage, bacterial burden was significantly reduced, relative to N2 (*Figure 5d and e*). These results indicate that AAIM-1 plays a role in limiting bacterial colonization, and its loss results in reduced survival due to hyper-colonization of the intestinal lumen.

## Fitness of *aaim-1* animals is dependent upon microbial environment

To investigate how *aaim-1* alleles can influence population structure, we set up competitive fitness assays. A *C. elegans* strain with a fluorescent marker (RFP::ZNFX-1) was co-plated with N2 or *aaim-1* mutants on *E. coli* OP50-1, *N. parisii* or *P. aeruginosa* PA14. Animals were grown for 8 days, such that the population was composed of adult F1s and developing F2s. On *E. coli* OP50-1, there is equal representation of N2 and *aaim-1* mutants in the population (*Figure 6a*). This is consistent with *aaim-1* mutants not having a developmental delay (*Figure 1b*) or a decrease in longevity (*Figure 5—figure supplement 2b*). In contrast, growth on *N. parisii* resulted in *aaim-1* alleles outcompeting the N2 strain. Conversely, *aaim-1* mutants on *P. aeruginosa* PA14 did significantly worse, being underrepresented in the population compared to N2 (*Figure 6a*). Interestingly, wild isolates of *C. elegans* do not carry any obvious loss of function alleles of *aaim-1* suggesting that natural conditions have selected for its retention (*Figure 6—figure supplement 1*; *Cook et al., 2017*).

Given the opposing fates of *aaim-1* mutants on *N. parisii* and *P. aeruginosa,* we investigated the effects of co-infection. Animals were infected with a maximal dose of *N. parisii* for 3 hr, prior to placement on lawns of PA14. For infections with a single pathogen, we observed similar results as before whereby *aaim-1* mutants have increased fitness in the presence of *N. parisii* and display lower levels of parasite burden but have increased bacterial accumulation when grown on PA14. In the presence of both pathogens, populations of *aaim-1* mutants display fewer gravid adults and increased amounts of *N. parisii* spores. (*Figure 6b and c*). These results suggests that coinfection with *N. parisii* and *P. aeruginosa* has synergistically negative effects on the fitness of *C. elegans*.

## Discussion

To identify host factors needed for microsporidia infection, we isolated mutants from a forward genetic screen that have a fitness advantage when challenged with *N. parisii* infection. This screen identified mutants in the poorly understood protein AAIM-1 (previously T14E8.4). Here, we demonstrate that this protein both promotes microsporidia invasion and limits colonization by pathogenic bacteria. Although we were unable to visualize the localization of secreted AAIM-1, our genetic and infection experiments strongly suggest that this protein acts in the intestinal lumen where both microsporidia invasion and bacterial colonization take place. The key role that AAIM-1 plays in immunity is further exemplified by its transcriptional regulation in response to infection (*Figure 7*).

The processes by which microsporidia invade host cells are poorly understood. We show that *N. parisii* spores are often angled in wild-type *C. elegans*, suggesting that successful invasion requires a particular spore orientation. In the absence of AAIM-1, spores are more often parallel to the intestinal lumen, where spores may fire without the successful deposition of the sporoplasm inside an intestinal cell. In contrast to previously described host and microsporidia proteins involved in invasion, AAIM-1 does not appear to be involved in promoting adhesion to the surface of host cells (*Han et al., 2020*;

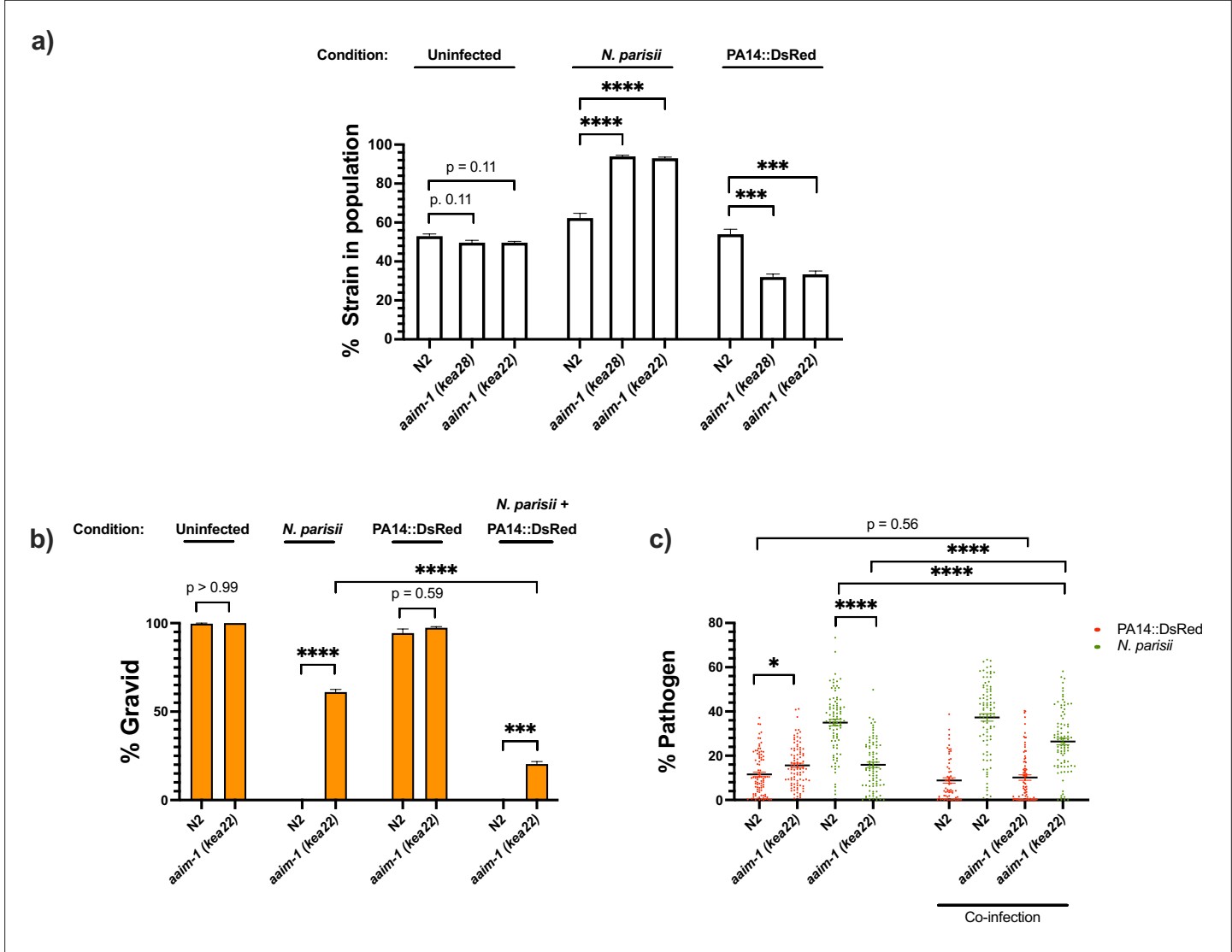

**Figure 6.** *aaim-1* alleles display enhanced fitness on *N. parisii*, but reduced fitness on *P. aeruginosa*. (**a**) Competitive fitness assays performed with a fluorescently marked strain (RFP::ZNFX1) mixed with either N2 or *aaim-1* mutants. These mixed populations of animals were plated at the L1 stage on either *E. coli*, a medium-2 dose of *N. parisii*, or on *P. aeruginosa*. After 8 days, the fraction of animals that did not display fluorescent germ granules was counted. Data is from three independent replicates of 20–270 worms each. (**b,c**) L1 stage N2 and *aaim-1* animals were either uninfected or infected with a maximal dose of *N. parisii*. These infected and uninfected population of animals were then washed and placed on either *E. coli* or PA14::DsRed. After 69 hr, animals were fixed and stained with DY96. Data is from three independent replicates of 60–150 worms each. (**b**) Graph displays percentage of gravid adults. (**c**) Quantified amount of either *N. parisii* (DY96) or *P. aeruginosa* (PA14::DsRed). Twelve to 30 worms were quantified per replicate. Mean ± SEM represented by horizontal bars. p-Values determined via one-way ANOVA with post hoc. Significance defined as ** $p < 0.01$, *** $p < 0.001$, **** $p < 0.0001$.

The online version of this article includes the following source data and figure supplement(s) for figure 6:

**Source data 1.** *aaim-1* alleles display enhanced fitness on *N. parisii*, but reduced fitness on *P. aeruginosa*.

**Figure supplement 1.** List of naturally occurring *aaim-1(T14E8.4)* variants in wild isolates of *C. elegans*.

**Figure supplement 1—source data 1.** Editable version of *Figure 6—figure supplement 1*.

*Tamim El Jarkass and Reinke, 2020*). Instead, AAIM-1 ensures an adequate distance of spores from the intestinal membrane, possibly allowing spores to be able to properly orient themselves to ensure successful host cell invasion. *N. parisii* spores are ~2.2 µm long by ~0.8 µm wide and the average width of the intestinal lumen at the L1 stage is ~0.6 µm (*Zhang et al., 2016*). Therefore, at the L1 stage spores may not be able to move freely, but at the L3 stage, where AAIM-1 is not needed for

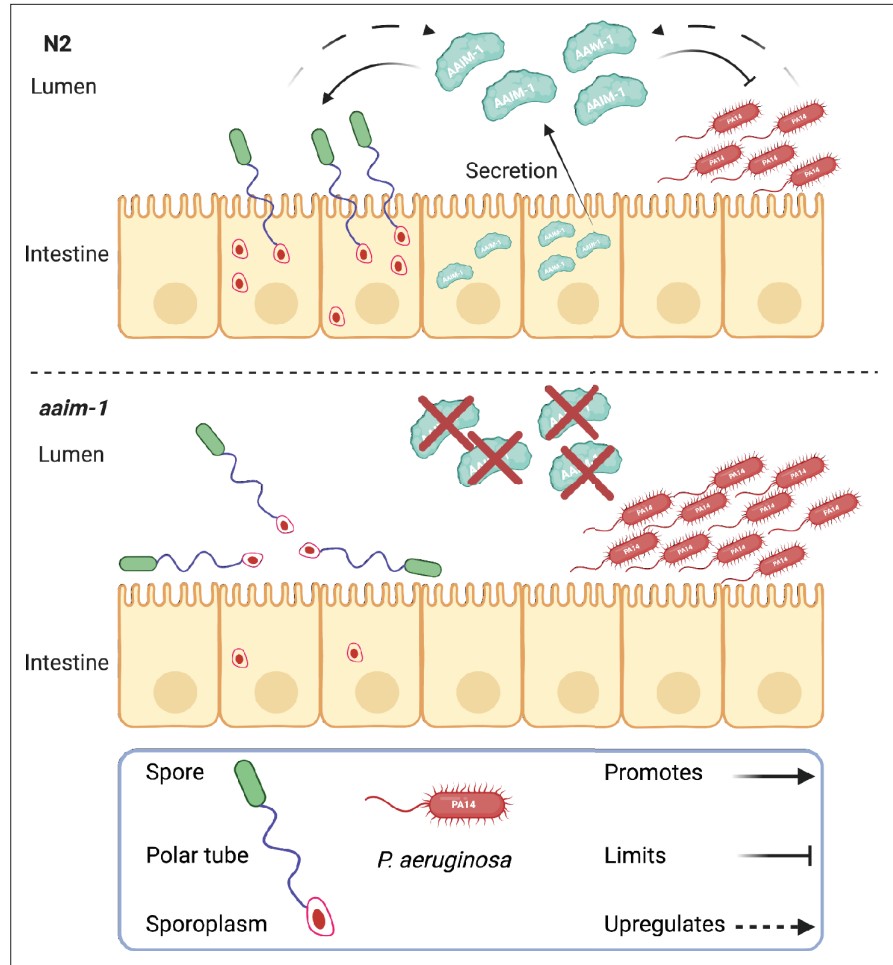

**Figure 7.** Secreted AAIM-1 functions in the intestinal lumen to limit bacterial colonization but is exploited by microsporidia to ensure successful invasion of intestinal cells. AAIM-1 is secreted from intestinal cells, where the protein limits bacterial colonization in the lumen. Additionally, AAIM-1 is parasitized by *N. parisii* spores to ensure successful orientation and firing during intestinal cell invasion. Infection by either of these two pathogens results in the upregulation of AAIM-1. Figure made with Biorender.com.

invasion, there is less of a constraint on spore movement as the luminal width increases to ~1.3 μm. Alternatively, the developmentally restricted role of AAIM-1 could be due stage-specific expression of other factors that work along with AAIM-1 to promote microsporidia invasion. Together, our results highlight the power of studying microsporidia invasion in the context of a whole animal model.

Several lines of evidence suggest that AAIM-1 plays a role in protecting animals against *P. aeruginosa*. First, *aaim-1* is upregulated in the intestine in response to PA14 exposure. Second, overexpression of AAIM-1 significantly decreases PA14 burden in the intestine. Third, loss of *aaim-1* leads to enhanced susceptibility and increased PA14 colonization. Fourth, competition assays show reduced reproductive fitness of *aaim-1* mutants on PA14. The survival phenotype of *aaim-1* mutants (~78% survival compared to wild type) is modest compared to loss of the p38 MAPK pathway (~51% survival compared to wild type). However, in competitive fitness assays, *aaim-1* mutants are ~60% less represented than wild type in the F2 generation. Taken together, our data suggests that in addition to promoting microsporidia invasion, AAIM-1, at least in part, limits bacterial colonization and decreases susceptibility to *P. aeruginosa*.

*C. elegans* employs a variety of proteins to protect against bacterial infection. Many of these proteins belong to several classes of antimicrobial effectors used to eliminate and prevent colonization by pathogenic bacteria (*Dierking et al., 2016*), are upregulated upon infection, and predicted to be secreted (*Suh and Hutter, 2012*; *Gallotta et al., 2020*). One class of secreted proteins that is known to have immune functions and prevent bacterial adherence are the mucins. These large,

glycosylated secreted proteins are upregulated during *C. elegans* infection and their knockdown alters susceptibility to *P. aeruginosa* infection. (*Strzyz, 2019*; *Hoffman et al., 2020*) AAIM-1 has many predicted mucin-like O-glycosylation sites on serine and threonine residues (*Steentoft et al., 2013*; *Jensen et al., 2010*; *Tran and Ten Hagen, 2013*). Thus, one possibility is that AAIM-1 may be functionally analogous to mucins, preventing the adhesion of microbes to the surface of intestinal cells. As AAIM-1 does not contain any known or conserved domains, further work will be necessary to determine its exact biochemical function.

C. elegans lives in a microbially dense environment containing a wide variety pathogens that *C. elegans* has evolved immunity towards (*Zhang et al., 2016*; *Schulenburg and Félix, 2017*; *Samuel et al., 2016*; *Sinha et al., 2012*; *Ashe et al., 2013*). Although loss of *aaim-1* provides a fitness advantage to *C. elegans* when grown in the presence of microsporidia, obvious loss of function alleles are not present in wild isolates sequenced thus far. Additionally, *aaim-1* mutants do not have observable defects when grown on non-pathogenic *E. coli*. This is in contrast to mutations in *pals-22* or *lin-35*, which negatively regulate the transcriptional response to infection and provide resistance to microsporidia infection when mutated, but at the cost of reduced reproductive fitness (*Willis et al., 2021*; *Reddy et al., 2017*). Loss of *aaim-1* disadvantages *C. elegans* when grown on *P. aeruginosa*, but not *S. aureus,* suggesting AAIM-1 does not broadly promote resistance to all bacterial pathogens. These findings demonstrate that there is a trade-off in host defense between microsporidia and some pathogenic bacteria. The opposing functions of *aaim-1* with different pathogens adds to the limited set of known examples of trade-offs that constrain the evolution of host defense to multiple biotic threats (*Toor and Best, 2016*; *Thaler et al., 1999*).

## Materials and methods
### Strain maintenance
*C. elegans* strains were grown at 21°C on nematode growth media (NGM) plates seeded with 10 x saturated *Escherichia coli* OP50-1 (*Willis et al., 2021*). Strains used in this study are listed in the key resources table (Appendix 1). For all infection assays, 15–20 L4 staged animals were picked onto 10-cm seeded NGM plates 4 days prior to sodium hypchlorite/1 M NaOH treatment. After 4 days, heavily populated non-starved plates were washed off with 1 ml M9, treated twice with 1 ml of sodium hypochlorite/1 M NaOH solution, and washed three times in 1 ml M9. Embryos were then resuspended in 5 ml of M9 and left to rock overnight at 21 °C. L1s were used in subsequent experiments no later than 20 hr after bleach treatment. All centrifugation steps with live animals/embryos were performed in microcentrifuge tubes at 845 x g for 30 s.

Throughout the paper, L1 refers to the stage immediately post hatching or bleach synchronization, L3 refers to 24 hr and L4 refers to 48 hr post plating of bleach synchronized L1s at 21 °C. L3 and L4 stage animals were washed off plates in M9 +0.1% Tween-20, followed by an additional wash to remove residual bacteria before infection with microsporidia, or plating on PA14.

### Forward genetic screen
6,000 L4 N2 hermaphrodites were mutagenized with a combination of 50 mM EMS and 85.4 mM ENU for 4 hr to achieve a large diversity of mutations within the genome (*Thompson et al., 2013*) P0 animals were then split and placed onto 48 10-cm NGM plates, F1s bleached and resulting F2s pooled onto five separate plates. 180,000 L1 F2 animals were plated onto a 10-cm plate with 10 million *N. parisii* spores and 1 ml 10 x saturated OP50-1. Animals were grown for 72 hr, to select for animals that display a fitness advantage phenotype with respect to N2. Each population was bleached and grown in the absence of infection for one generation, in order to prevent the effects of intergenerational immunity (*Willis et al., 2021*). Two more cycles of infection followed by growing worms in the absence of infection was performed. Populations of bleached L1s were then infected with either 20 or 40 million spores and grown for 76 hr. Worms were then washed into 1.5 ml microcentrifuge tubes and 1 ml of stain solution (1 x PBS/0.1% Tween-20/2.5 mg/ml DY96/1% SDS) was added. Samples were incubated with rotation for 3.5 hr and then washed three times with M9 +0.1% Tween-20. Individual worms that had embryos, but not spores, were picked to individual plates. Each of the four *fawn* strains was isolated from a different mutant pool.

## Whole genome sequencing

N2 and *fawn* isolates were each grown on a 10-cm plate until all *E. coli* was consumed. Each strain was washed off with M9 and frozen at –80 °C. DNA was extracted using Gentra puregene Tissue Kit (QIAGEN). Samples were sequenced on an Illumina HiSeq 4000, using 100 base paired end reads.

## MIP-map

Molecular inversion probes were used to map the underlying causal mutations in *fawn* isolates as previously described (*Mok et al., 2017*). Briefly, *fawn* hermaphrodites were crossed to males of the mapping strain DM7448 (VC20019 Ex[*pmyo3::*YFP*]*) hereafter referred to as VC20019 (*Mok et al., 2020*). Next, 20 F1 hermaphrodite cross progeny, identified as those carrying *pmyo3*::YFP, were isolated and allowed to self. F2s were then bleached, and 2500 L1s were exposed to a medium-2 dose of *N. parisii* spores representing the first round of selection. Two plates of 2500 F3 L1s were set up. The experimental plate was grown in the absence of infection for one generation, to negate intergenerational immunity (*Willis et al., 2021*). A second plate of 2500 L1s was allowed to grow for 72 hr and then frozen in H$_2$O at –80 °C, until used for genomic preparation. The selection and rest steps were repeated once more, and a second frozen sample of worms was taken at the end of the mapping experiment. This process was also performed for a cross between N2 hermaphrodites and males of the mapping strain VC20019, as a negative control to identify non-causal loci that may be selected for reasons other than resistance to infection. Two genomic preparations, corresponding to the two rounds of selection, were used as template for MIP capture, to generate multiplexed libraries for sequencing. An Illumina Mini-seq was used to generate sequencing data that was subjected to demultiplexing via R, and selection intervals were defined as those immediately adjacent to the region on the chromosome carrying the fewest proportion of reads corresponding to the mapping strain, VC20019. This interval was then used to scan for putative causal alleles, resulting in the identification of the four *aaim-1* alleles in the four *fawn* isolates.

## Identification of causal gene

Variants were identified using a BWA-GATK pipeline. Briefly, sequencing reads were checked for sequence quality using FastQC (http://www.bioinformatics.babraham.ac.uk/projects/fastqc/) and bases lower than a quality threshold of 30 were trimmed off with Trimmomatic using a sliding window of 4 bases and minimum length of 36 bases (*Bolger et al., 2014*). Reads were aligned to the *C. elegans* N2 reference genome (release W220) using BWA-mem (*Li and Durbin, 2009*). Alignments were sorted by coordinate order and duplicate reads removed using Picard (*Broad Institute, 2019*). Prior to variant calling, reads were processed in Genome Analysis Tool Kit (GATK) v3.8.1 (*DePristo et al., 2011*), to perform indel realignment and base quality score recalibration using known *C. elegans* variants from dbSNP, build 138 (http://www.ncbi.nlm.nih.gov/SNP/). GATK HaplotypeCaller was used to call variants, and results were filtered for a phred-scaled Qscore >30 and to remove common variants found previously in multiple independent studies. Finally, Annovar (*Wang et al., 2010*) was used to obtain a list of annotated exonic variants for each sequenced strain.

## Microsporidia infection assays

*N. parisii* (ERTm1) and *N. ausubeli* (ERTm2) spores were prepared as described previously (*Willis et al., 2021*). All infections were carried out on 6-cm NGM plates, unless otherwise specified by spore dose (see *Supplementary file 1*), or experimental method. 1000 bleach-synchronized L1s were added into a microcentrifuge tube containing 400 µl of 10 X *E. coli* OP50-1, and spores. After pipetting up and down, this mixture was top plated onto an unseeded 6-cm NGM plate, and left to dry in a clean cabinet, prior to incubation at 21 °C for 72 hr. Infections set up on 3.5-cm plates used 160 µl of 10 x *E. coli* OP50-1 and 400 L1s.

## Infection of embryos hatched on plates

Twenty-five 72-hr-old synchronized animals of each strain were picked onto 3.5-cm unseeded NGM plates seeded with 16 µl of 10 x *E. coli* OP50-1. Plates were incubated at 21 °C for 2 hr. Adults were then picked off, and a mixture of 144 µl of 10 x *E. coli* OP50-1 and a low dose of *N. parisii* spores were added to each plate. Animals were fixed and stained after 72 hr.

## Pulse-chase infection assay

A total of 6000 bleach synchronized animals were exposed to a medium-1 (*Figure 3—figure supplement 1*) or medium-3 (*Figure 3*) dose of spores, 10 µl of 10 x *E. coli* OP50-1 in a total volume of 400 µl made up with M9. To assay pathogen clearance 3 hpi, animals were washed off in 1 ml M9 +0.1% Tween-20, and split into two populations. The first half was fixed with acetone to represent initial infectious load, while the other half was washed twice in M9 +0.1% Tween-20 to remove residual spores in the supernatant and prevent additional infection from occurring. These washed worms were then plated on 6-cm unseeded NGM plates with 40 µl 10 x OP50-1, and 360 µl M9 and left to incubate at 21 °C for 21 additional hours before fixation.

## Spore localization and firing assays

Strains were infected as described for the pulse infection assays for either 45 min or 3 hr. Animals were then washed off plates, fixed, and stained with DY96 and an *N. parisii* 18 S RNA FISH probe. FISH⁺ DY96⁺ events represent unfired spores, FISH⁻ DY96⁺ events represent fired spores, and FISH⁺ DY96⁻ events represent sporoplasms. Percentage of fired spores is defined as the number of FISH⁻ DY96⁺ events over the total number of spores (DY96⁺).

To assess spore orientation, the localization of calcofluor white spores relative to the apical membrane of the intestine was measured in live anaesthetized animals using differential interference contrast microscopy. To determine if a spore was angled, straight lines were extended from both ends of the spore independently. If either of these two lines crossed the apical membrane, a spore was considered angled. If not, the spore was considered parallel. Distance of spores from the apical membrane was assessed by measuring perpendicular distance from the central edge of a parallel spore to the apical membrane. All measurements were performed with FIJI (*Schindelin et al., 2012*) using the angle tool or the straight line tool respectively, followed by the Analyze → measure option. Images in *Figure 4a* were taken in N2 and *aaim-1 (kea22)* animals carrying PGP-1::GFP (*Sato et al., 2007*) to label the apical intestinal membrane, thus outlining the lumen.

## Intestinal lumen measurements

Measurements were performed on live anaesthetized worms used for spore localization assays (see above). The width of the lumen was determined by extending a straight line from the apical membrane on one end of the worm to that directly across on the other end, at the midpoint of the intestine, and the distance measured in FIJI, via the straight line tool followed by the Analyze → measure option.

## Fixation

Worms were washed off infection plates with 700 µl M9 +0.1% Tween-20 and washed once in 1 ml M9 +0.1% Tween-20. All microsporidia infected samples were fixed in 700 µl of acetone for 2 min at room temperature prior to staining. All *P. aeruginosa* PA14::DsRed infected samples, as well as competitive fitness assays involving RFP::ZNFX-1 were fixed in 500 µl of 4% paraformaldehyde (PFA) for 30 min at room temperature prior to mounting on slides.

## Live imaging

Animals were mounted on 2% Agarose pads in 10 µl of 10–25 mM Sodium Azide. This technique was used for spore localization assays, transcriptional reporter imaging, and assessing PA14::DsRed colonization in transgenic animals.

## Chitin staining

The chitin binding dye Direct yellow 96 (DY96) was used to assess host fitness (gravidity) as well as parasite burden. A total of 500 µl of DY96 solution (1 x PBST, 0.1% SDS, 20 µg/ml DY96) was added to washed worm pellets and left to rock for 20–30 min at room temperature. Worms were then resuspended in 20 µl of EverBrite Mounting Medium (Biotium), and 10 µl mounted on glass slides for imaging.

To prestain spores prior to infection, 0.5 µl of calcofluor white solution (CFW) (Sigma- Aldrich 18909) was added per 50 µl of spores, pipetted up and down gently and left for 2 min at room temperature prior to infection.

## FISH staining

To quantify the number of sporoplasms in *N. parisii* infected animals, the MicroB FISH probe (ctct cggcactccttcctg) labelling *N. parisii* 18 S RNA was used. Animals were fixed in acetone, washed twice in 1 ml PBST, and once in 1 ml of hybridization buffer (0.01% SDS, 900 mM NaCl, 20 mM TRIS pH 8.0). Samples were then incubated overnight in the dark at 46 °C with 100 µl of hybridization buffer containing 5 ng/µl of the MicroB FISH probe conjugated to Cal Fluor 610 (LGC Biosearch Technologies). Samples were then washed in 1 ml of wash buffer (Hybridization buffer + 5 mM EDTA), followed by incubation with 500 µl wash buffer at 46 °C in the dark. To visualize sporoplasms and spores simultaneously, the final incubation was replaced with 500 µl DY96 solution and incubated in the dark at room temperature prior to resuspension in 20 µl of EverBrite Mounting Medium (Biotium).

## Microscopy and image quantification

All imaging was performed using an Axio Imager.M2 (Zeiss), except for images of the transcriptional reporter in *Figure 5—figure supplement 1*, which were generated using an Axio Zoom V.16 (Zeiss) at a magnification of 45.5 x. Images were captured via Zen software and quantified under identical exposure times per experiment. Gravidity is defined as the presence of at least one embryo per worm, and animals were considered infected by 72 hr if clumps of spores were visible in the body of animals as seen by DY96. FISH-stained animals were considered infected if at least one sporoplasm was visible in intestinal cells.

To quantify fluorescence within animals (pathogen burden, bead accumulation, and GFP), regions of interest were used to outline every individual worm from anterior to posterior, unless otherwise specified in the methods. Individual worm fluorescence from variable assays (GFP or DsRed) were subjected to the 'threshold' followed by 'measure' tools in FIJI (*Schindelin et al., 2012*). To assess PA14::DsRed burden in transgenic animals, regions of interest were generated from the beginning of the intestines (int1) to the posterior end of the worm to prevent the p*myo2*::mCherry co-injection marker signal from interfering with quantifications. When assessing pathogen burden in gravid animals stained with DY96, thresholding was used to quantify spore signal without including signal from embryos.

## *Pseudomonas aeruginosa* infection experiments

For all *Pseudomonas* assays, a single colony was picked into 3 ml of LB and grown overnight at 37 °C, 220 rpm for 16–18 hr. 10 µl (for 3.5-cm plate) or 50 µl (for 6-cm plate) of culture was spread onto slow killing (SK) plates to form a full lawn, except in the case of competitive fitness assays (see below). Seeded plates were placed at 37 °C for 24 hr, followed by 25 °C for 24 hr prior to use. Plates were seeded fresh prior to each experiment. To assess colonization, 1000 synchronized animals were grown on PA14::DsRed for either 24 or 48 hr at 25 °C. Animals were washed off with 1 ml M9 +0.1% Tween-20, and washed twice thereafter, prior to fixation.

To quantify survival of individual strains on PA14, 3.5-cm SK plates were seeded with 10 µl of PA14::DsRed,(*Dunn et al., 2006*) to form full lawns. 60 L4s were picked onto each of three, 3.5- cm plates per strain, and 24 hr later, 30 animals from each were picked onto a new 3.5-cm plate. Survival was monitored from 24 hr post L4, three times per day. Survival was assessed based on response to touch. Carcasses were removed, and surviving animals were placed onto fresh 3.5-cm plates every 24 hr. Animals were grown at 25 °C for the duration of the experiment. Technical triplicate data was pooled to represent a single biological replicate. The experiment was carried out until no more worms had survived. Survival curves were generated via GraphPad Prism 9.0, and the Log rank (mantel-cox) test was used to generate P-values. $TD_{50}$ values were calculated as previously described,(*Styer et al., 2008*) utilizing GraphPad Prism 9.0 and applying a non-linear regression analysis on survival curves.

## *Staphylococcus aureus* infection experiments

3.5-cm tryptic soy agar (TSA) plates supplemented with 10 µg/ml nalidixic acid (Nal) and seeded with *S. aureus* NCTC8325 (*Sifri et al., 2003*) were utilized and survival quantified as described previously (*Powell and Ausubel, 2008*). Briefly, a 1:10 dilution of an overnight *S. aureus* culture was utilized to seed 3.5-cm TSA + Nal plates, incubated at 37 °C for 3 hr and stored at 4 °C overnight. Thirty L4s were picked onto three TSA + Nal plates per strain, and survival quantified three times a day until all animals were dead. Animals were transferred to new seeded TSA + Nal plates every 24 hr. Survival was

assessed based on response to touch. Technical triplicate data was pooled to represent a single biological replicate. Survival curves were generated via GraphPad Prism 9.0, and the Log rank (mantel-cox) test was used to generate p-values.

## Transgenic strain construction

N2 or *aaim-1 (kea22)* animals were injected with a 100 ng/µl injection mix composed of 50 ng/µl of template, 5 ng/µl of p*myo2*::mCherry, and 45 ng/µl of pBSK (*Ponchon et al., 2009*). Three independent lines were generated for each injected construct.

Gateway BP cloning (*Walhout et al., 2000*; *Hartley et al., 2000*) was performed to insert AAIM-1 and GFP into pDONR221. Around the horn PCR, (*Moore and Prevelige, 2002*) was used to insert a 3 x Flag sequence at the C-terminus of this construct. Gibson assembly was used to generate different tissue specific clones driving *aaim-1* expression. P*aaim-1, aaim-1* and p*spp-5* were cloned from N2 genomic DNA, *pmyo2* was cloned from pCFJ90 (*Frøkjaer-Jensen et al., 2008*). GFP and 3 x Flag sequences were cloned from pDD282 (*Frøkjaer-Jensen et al., 2008*). *SPΔaaim-1*was amplified from *aaim-1*::3xFlag in pDONR221 by omitting the first 17 amino acids, the putative secretion signal as predicted via SignalP 5.0 (*Almagro Armenteros et al., 2019*). All clones possessed an *unc-54* 3' UTR. See key resources table for primer sequences (Appendix 1).

## CRISPR-Cas9 mutagenesis

To generate a deletion allele of *aaim-1* via CRISPR-Cas9 mutagenesis, steps were taken as described here (*Dickinson et al., 2015*). Briefly, 2 crRNA's were designed using CRISPOR (*Concordet and Haeussler, 2018*), (*Dokshin et al., 2018*), near the start and stop sites of *aaim-1* and generated via IDT. A repair template was designed to contain 35 base pairs of homology upstream and downstream of the cut sites. *Streptococcus pyogenes* Cas9 3NLS (10 µg/µl) IDT and tracrRNA (IDT #1072532) were utilized. Reaction mixes were prepared as described previously. pRF4 (*Mello et al., 1991*) was co-injected with the Cas9 ribonucleoprotein, and F1 rollers picked. Deletions were identified via PCR primers situated outside the cut sites.

## Bead-feeding assays

1,000 synchronized L1 animals were mixed with 0.2 µm green fluorescent polystyrene beads (Degradex Phosphorex) at a ratio of 25:1 in a final volume of 400 µl containing 10 µl of 10 x *E. coli* OP50-1, 16 µl of beads and up to 400 µl with M9. Animals were incubated with beads for 3 hr, washed off with M9 +0.1% Tween-20 and fixed with 4% PFA for 30 min at room temperature. Bead accumulation was measured as a percentage of the total animal exhibiting fluorescent signal, using FIJI.

## Lifespan assays

Lifespan assays were performed as described previously (*Amrit et al., 2014*). In brief, 120 synchronized L4 animals were utilized per strain, with every 15 animals placed on a single 3.5-cm NGM plate (A total of 8 plates, with 15 animals each per strain). Animals were transferred to a new seeded 3.5-cm NGM plate every 2 days, for a total of 8 days (four transfers), ensuring no progeny were transferred alongside adults. After day 8, survival was quantified daily, on the same plate, via response to touch. Any animals that exhibited internal hatching, protruding intestines, or were found desiccated on the edges of the plate were censored. Survival curves were generated via GraphPad Prism 9, and the Log rank (mantel-cox) test was used to generate p values.

## Immunofluorescence (IF)

IF was performed as described previously (*Crittenden and Kimble, 2009*); however, all steps post-dissection were performed in microcentrifuge tubes, and intestines were pelleted on a mini tabletop microcentrifuge for a few seconds. Briefly, animals were dissected to extrude intestinal tissue. Two 25-mm gauge needles on syringes were used to create an incision near the head and/or tail of the animals. Dissections were performed in 5 µl of 10 mM levamisole on glass slides to encourage intestinal protrusion. Fixation, permeabilization and blocking was performed as described previously (*Crittenden and Kimble, 2009*). Primary M2 anti-Flag antibody (Sigma F1804) was used at 1:250 overnight at 4 °C, and secondary goat anti-mouse Alexa fluor 594 (Thermo Fisher A32742) at 1:300 for 1 hr

at room temperature. Animals were mounted in 20 µl of EverBrite Mounting Medium (Biotium) and placed on glass slides for imaging.

## Competitive fitness assays

N2 or *aaim-1* mutants were grown together with RFP::Znfx1 YY1446 *(gg634)*, which labels the germ granules and can be observed in all developmental stages (*Wan et al., 2018*). For *N. parisii* infections, 10-cm NGM plates were seeded with 1 ml of 10 x OP50-1 and a medium-2 dose of spores (no spores were used for uninfected plates). 10 L1s from each strain were picked onto lawns of spores and *E. coli* OP50-1 immediately after drying, and grown for 8 days at 21 °C, washed off with M9 +0.1% Tween-20, and fixed. For *P. aeruginosa* infections, 3.5-cm SK plates were seeded with a single spot of 20 µl of PA14 in the center of the plate. 10 L1s of each strain were placed on plates and grown at 21 °C for 8 days and then washed off with M9 +0.1% Tween-20. The percentage of animals that did not display RFP germ granules (i.e. N2 or *aaim-1* mutants) was determined by quantifying all animals on the plate, including F1 adults and L1/L2 stage F2 animals.

## Co- infections with *N. parisii* and *P. aeruginosa*

Co-infection assays were performed by first pulse infecting co-infection and *N. parisii* single infection groups with a maximal dose of spores for 3 hr on unseeded 6-cm NGM plates as described above. PA14::DsRed single infections were pulsed with a volume of M9 to match that of the spores. Animals were then washed off in 1 ml of M9 +0.1% Tween-20, followed by two more washes, prior to placement on full lawns of PA14::DsRed on a 6-cm SK plates prepared as described above. *N. parisii* single infections were placed on a 6-cm NGM plate pre-seeded with 200 µl of 10 x OP50-1. Plates were incubated at 21 °C.

## Phylogenetic analysis

Homology between AAIM-1 and other proteins was determined with protein BLAST (https://blast.ncbi.nlm.nih.gov/Blast.cgi) using default parameters. Sequences with an E-value of at least 10−5 were aligned using MUSCLE (https://www.ebi.ac.uk/Tools/msa/muscle/) using default parameters. Phylogenetic tree of homologs was generated using RAxML BlackBox (https://raxml-ng.vital-it.ch/) using default parameters and 100 boot straps. Tree was visualized using FigTree v1.4.4 (http://tree.bio.ed.ac.uk/software/figtree/).

## Statistical analysis

All data analysis was performed using GraphPad Prism 9.0. One-way Anova with post hoc (Tukey test) was used for all experiments unless otherwise specified in figure legends. Statistical significance was defined as $p < 0.05$.

## Acknowledgements

We thank Ashley M Campbell, Alexandra R Willis, and Kristina Sztanko for providing helpful comments on the manuscript. This work was supported by the Canadian Institutes of Health Research grant no. 400,784 and an Alfred P Sloan Research Fellowship FG2019-12040 (to AWR). This work was supported by National Institutes of Health (http://www.nih.gov/) under R01 AG052622 and GM114139 to ERT. Some strains were provided by the CGC, which is funded by NIH Office of Research Infrastructure Programs (P40 OD010440) and we thank WormBase.

## Additional information

### Funding

| Funder | Grant reference number | Author |
|---|---|---|
| Canadian Institutes of Health Research | 400784 | Aaron W Reinke |
| Alfred P. Sloan Foundation | FG2019-12040 | Aaron W Reinke |

| Funder | Grant reference number | Author |
|---|---|---|
| National Institutes of Health | AG052622 | Emily R Troemel |
| National Institutes of Health | GM114139 | Emily R Troemel |

The funders had no role in study design, data collection and interpretation, or the decision to submit the work for publication.

## Author contributions

Hala Tamim El Jarkass, Conceptualization, Data curation, Formal analysis, Investigation, Methodology, Validation, Visualization, Writing – original draft, Writing – review and editing; Calvin Mok, Michael R Schertzberg, Formal analysis, Writing – review and editing; Andrew G Fraser, Funding acquisition, Supervision; Emily R Troemel, Funding acquisition, Resources, Supervision, Writing – review and editing; Aaron W Reinke, Conceptualization, Data curation, Funding acquisition, Investigation, Methodology, Project administration, Resources, Supervision, Writing – original draft, Writing – review and editing

## Author ORCIDs

Hala Tamim El Jarkass http://orcid.org/0000-0003-0118-5978
Andrew G Fraser http://orcid.org/0000-0001-9939-6014
Emily R Troemel http://orcid.org/0000-0003-2422-0473
Aaron W Reinke http://orcid.org/0000-0001-7612-5342

## Decision letter and Author response

Decision letter https://doi.org/10.7554/eLife.72458.sa1
Author response https://doi.org/10.7554/eLife.72458.sa2

## Additional files

### Supplementary files

• Supplementary file 1. Spore doses utilized in this study. The different species of microsporidia used in the study, and the varying doses (as defined in figure legends) are listed. Plate concentration refers to the number of spores occupied per $cm^2$ of either 3.5, 6, or 10 cm NGM plates (indicated by *, ** or *** in the 'total spores on assay plate' column). The total number of spores present on a single assay plate are listed for the various doses in millions of spores.

• Transparent reporting form

### Data availability

All data generated during this study have been uploaded as source data files for each figure.

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

# Appendix 1

## Appendix 1—key resources table

| Reagent type (species) or resource | Designation | Source or reference | Identifiers | Additional information |
|---|---|---|---|---|
| Gene (*Caenorhabditis elegans*) | *aaim-1* | This paper | *T14E8.4* | Wormbase ID: WBGene00043981 |
| Genetic reagent (*Caenorhabditis elegans*) | N2 | *Caenorhabditis* genetic center (CGC) | N2 RRID:SCR_007341 | Wild-type, Bristol strain. |
| Genetic reagent (*Caenorhabditis elegans*) | *fawn-1* (AWR 05) | This paper | *aaim-1 (kea89) X* | C127T, Q43Stop |
| Genetic reagent (*Caenorhabditis elegans*) | *fawn-2* (AWR 11) | This paper | *aaim-1 (kea28) X* | 2.2 kb deletion |
| Genetic reagent (*Caenorhabditis elegans*) | *fawn-3* (AWR 17) | This paper | *aaim-1 (kea91) X* | G221A splice site mutation |
| Genetic reagent (*Caenorhabditis elegans*) | *fawn-4* (AWR 03) | This paper | *aaim-1 (kea103) X* | C1286T, A429V |
| Genetic reagent (*Caenorhabditis elegans*) | DM7748 | **Mok et al., 2020** DOI: 10.1534/g3.120.401656. | VC20019 (Ex[*Pmyo-3::YFP*]) | Mapping strain |
| Genetic reagent (*Caenorhabditis elegans*) | RB563 | *Caenorhabditis* genetic center (CGC) | *aaim-1 (ok295) X* | Large gene deletion in *aaim-1* and neighbouring gene *dop-3* |
| Genetic reagent (*Caenorhabditis elegans*) | AWR 73 | This paper | *aaim-1 (kea22) X* | 3 x outcrossed CRISPR-Cas9 generated deletion allele of *aaim-1*. |
| Genetic reagent (*Caenorhabditis elegans*) | AWR 83 | This paper | *aaim-1 (kea28) X* | 6 x outcrossed *fawn-2 (kea28)* |
| Genetic reagent (*Caenorhabditis elegans*) | DA465 | *Caenorhabditis* genetic center (CGC) | *eat-2 (ad465) II* | Feeding defective mutant. |
| Genetic reagent (*Caenorhabditis elegans*) | AWR 131 | This paper | N2 Ex[p*myo2::mCherry::Unc54*, p*aaim-1::GFP::3xFlag::Unc54*] | *aaim-1* transcriptional reporter in N2 background with a pharyngeal mCherry co-injection marker. |
| Genetic reagent (*Caenorhabditis elegans*) | AWR 125 | This paper | *aaim-1(kea22* Ex[p*myo2::mCherry::Unc54*, p*aaim-1::GFP::3xFlag::Unc54*]) | *aaim-1* transcriptional reporter in *aaim-1 (kea22)* mutant background with a pharyngeal mCherry co-injection marker. |
| Genetic reagent (*Caenorhabditis elegans*) | AWR 122 | This paper | *aaim-1(kea22* Ex[p*myo3::mCherry::Unc54*, p*aaim-1::GFP::Unc54*]) | *aaim-1* transcriptional reporter in *aaim-1 (kea22)* mutant background with a body wall muscle mCherry co-injection marker. |
| Genetic reagent (*Caenorhabditis elegans*) | AWR 115 | This paper | *aaim-1 (kea22* Ex[p*myo2::mCherry::Unc54*, p*aaim-1::aaim-1::Unc54*]) | *aaim-1* over expression in an *aaim-1 (kea22)* mutant background with a pharyngeal mCherry co-injection marker. |

*Appendix 1 Continued on next page*

*Appendix 1 Continued*

| Reagent type (species) or resource | Designation | Source or reference | Identifiers | Additional information |
|---|---|---|---|---|
| Genetic reagent (*Caenorhabditis elegans*) | AWR119 | This paper | *aaim-1 (kea22 Ex[pmyo2::mCherry::Unc54, paaim-1::aaim-1::3xFlag::Unc54])* | *aaim-1::3xFlag* over expression in an *aaim-1 (kea22)* mutant background with a pharyngeal mCherry co-injection marker. |
| Genetic reagent (*Caenorhabditis elegans*) | AWR 127 | This paper | *aaim-1 (kea22 Ex[pmyo2::mCherry::Unc54, paaim-1::SPΔ aaim-1::3xFlag::Unc54])* | Signal peptide mutant *aaim-1::3xFlag* over expression in an *aaim-1 (kea22)* mutant background with a pharyngeal mCherry co-injection marker. |
| Genetic reagent (*Caenorhabditis elegans*) | AWR129 | This paper | *aaim-1 (kea22 Ex[pmyo2::mCherry::Unc54, pspp-5:: aaim-1::3xFlag::Unc54])* | Intestinal *aaim-1::3xFlag* over expression in an *aaim-1 (kea22)* mutant background with a pharyngeal mCherry co-injection marker. |
| Genetic reagent (*Caenorhabditis elegans*) | YY1446 | *Caenorhabditis* genetic center (CGC) | *znfx-1(gg634[HA::tagRFP::znfx-1]) II.* | RFP germ granules |
| Genetic reagent (*Caenorhabditis elegans*) | GK288 | **Sato et al., 2007** DOI: 10.1038/nature05929 | *unc-119(ed3); dkIs166[popt-2::PGP-1::GFP, unc-119(+)]* | Apical intestinal membrane marker. |
| Genetic reagent (*Caenorhabditis elegans*) | KU25 | *Caenorhabditis* genetic center (CGC) | *pmk-1 (km25) IV* | p38 Map kinase loss of function mutant. |
| Genetic reagent (*Caenorhabditis elegans*) | AWR182 | This paper | *aaim-1 (kea22) X; dkIs166[popt-2::PGP-1::GFP, unc-119(+)] IV* | *aaim-1 (kea22)* mutant allele crossed into GK288. |
| Genetic reagent (*Escherichia coli*) | OP50-1 | *Caenorhabditis* genetic center (CGC) | OP50-1 | Uracil auxotroph. B strain. Streptomycin resistant. |
| Genetic reagent (*Escherichia coli*) | 5-alpha competent *E. coli* | New England Biolabs (NEB) | Cat#: C2987H | *E. coli* background in which transformations for molecular cloning were performed. |
| Genetic reagent (*Nematocida parisii*) | *N. parisii* (ERTm1) | This paper | *N. parisii* (ERTm1) | Nematode intestinal infecting species of microsporidia. |
| Genetic reagent (*Nematocida ausubeli*) | *N. ausubeli* (ERTm2) | This paper | *N. ausubeli* (ERTm2) | Nematode intestinal infecting species of microsporidia. |
| Genetic reagent (*Pseudomonas aeruginosa*) | PA14::DsRed | **Dunn et al., 2006** DOI: 10.1128/AEM.72.1.802–810.2006 | PA14::DsRed | DsRed labelled strain of PA14. |
| Genetic reagent (*Staphylococcus aureus*) | NCTC 8325 | **Sifri et al., 2003** DOI: 10.1128/IAI.71.4.2208–2217.2003 | NCTC 8325 | *S. aureus* isolate used for *C. elegans* killing assays. |
| Antibody | M2 anti Flag (Mouse monoclonal) | Sigma | Cat#: F1804 RRID:AB_262044 | IF (1:250) |
| Antibody | Anti mouse Alexafluor 594 (Goat polyclonal) | Thermo Fisher | Cat#: A32742 RRID:AB_2762825 | IF (1:300) |
| Sequence-based reagent | Reverse primer | Integrated DNA technologies (IDT) | Reverse primer to amplify *aaim-1* | 5'-ttaatttttttgctgg tgagg-3' |
| Sequence-based reagent | Forward primer | Integrated DNA technologies (IDT) | Forward primer to generate *SPΔaaim-1* | 5'-atgctaaaggatttct tgccgtg-3' |
| Sequence-based reagent | Forward primer | Integrated DNA technologies (IDT) | Forward primer to amplify p*aaim-1* | 5'-ttagtttggaaatgca caaaaaaactgatctct-3' |

*Appendix 1 Continued on next page*

*Appendix 1 Continued*

| Reagent type (species) or resource | Designation | Source or reference | Identifiers | Additional information |
|---|---|---|---|---|
| Sequence-based reagent | Reverse primer | Integrated DNA technologies (IDT) | Reverse primer to amplify p*aaim-1* | 5-cagtggacttctgctt attaaaatgacttc-3' |
| Sequence-based reagent | Forward primer | Integrated DNA technologies (IDT) | Forward primer to amplify p*myo2* | 5'-cattttatatctgagt agtatcctttgctttaaatg tcc-3' |
| Sequence-based reagent | Reverse primer | Integrated DNA technologies (IDT) | Reverse primer to amplify p*myo2* | 5'- gcatttctgtgtctga cgat-3' |
| Sequence-based reagent | Forward primer | Integrated DNA technologies (IDT) | Forward primer to amplify p*spp5* | 5'-aaagcaaaatatcatt atttgggaaaatc-3' |
| Sequence-based reagent | Reverse primer | Integrated DNA technologies (IDT) | Reverse primer to amplify p*spp5* | 5'-tctgtaataaaataaa ttgaaatgaaacac-3' |
| Sequence-based reagent | Forward primer | Integrated DNA technologies (IDT) | Forward primer to amplify GFP from pDD282 | 5'-atgagtaaagga gaagaattgttcact-3' |
| Sequence-based reagent | Reverse primer | Integrated DNA technologies (IDT) | Reverse primer to amplify GFP from pDD282 | 5'-ttacttgtagagctcg tccattccg-3' |
| Sequence-based reagent | Forward Ultramer | Integrated DNA technologies (IDT) | Forward ultramer to add a Gly Ala Gly Ser linker and 3 x Flag with *stop codon* to C-Terminal end of constructs in pDDONR221 via round the horn PCR.[5] | 5'-ggagccggatctgatt ataaagacgatgacga taagcgtgactacaag gacgacgacaca ag cgtgattacaaggatg acgatgacaagagata aacccagctttcttgtacaa agttg-3' |
| Sequence-based reagent | MicroB FISH probe conjugated to Cal Fluor 610 | LGC Biosearch Technologies | 18 s RNA FISH probe | 5'-ctctcggcactccttc ctg-3' |
| Sequence-based reagent | Alt-R CRISPR-Cas9 tracrRNA 5 nmol | Integrated DNA technologies (IDT) | Cat#: 1072532 | |
| Sequence-based reagent | 5' sgRNA | Integrated DNA technologies (IDT) | Guide RNA | 5'-aataaatggcataagt taag-3' |
| Sequence-based reagent | 3' sgRNA | Integrated DNA technologies (IDT) | Guide RNA | 5'-tttacaggcgtgtttc attg-3' |
| Recombinant protein | Alt-R S.p. Cas9 Nuclease V3, 100 ug | Integrated DNA technologies (IDT) | Cat#: 1081058 | |
| Recombinant protein | Phusion High-Fidelity DNA Polymerase | New England Biolabs (NEB) | Cat#: M0530L | DNA polymerase used for all molecular cloning steps. |
| Recombinant DNA reagent | pBSK | *Ponchon et al., 2009* DOI: 10.1038/nprot.2009.67. | pBSK RRID:Addgene_67504 | Addgene ID: 67,504 |
| Recombinant DNA reagent | Gateway pDONR221 | Invitrogen | Cat#: 12536017 | |
| Recombinant DNA reagent | pCFJ90 | *Frøkjaer-Jensen et al., 2008* DOI: 10.1038/ng.248. | pCFJ90 RRID:Addgene_19327 | Addgene ID: 19,327 |
| Recombinant DNA reagent | pDD282 | *Dickinson et al., 2015* DOI: 10.1534/genetics.115.178335. | pDD282 RRID:Addgene_66823 | Addgene ID: 66,823 |
| Commercial assay or kit | NEBuilder HiFi DNA assembly | New England biolabs (NEB) | Cat#: 3E2621 | Kit used for Gibson assembly. |
| Commercial assay or kit | Monarch PCR and DNA Cleanup Kit | New England biolabs (NEB) | Cat#: T1030S | PCR purification kit for amplicons used in downstream molecular cloning steps. |
| Commercial assay or kit | QIAprep spin miniprep kit | Qiagen | Cat#: 27,106 | Kit used for extraction of DNA from bacterial clones. |
| Chemical compound, drug | Direct yellow 96 | Sigma- Aldrich | Cat#: S472409-1G | Chitin binding dye. |
| Chemical compound, drug | Calcofluor white | Sigma- Aldrich | Cat#: 18,909 | Chitin binding dye. |

*Appendix 1 Continued*

| Reagent type (species) or resource | Designation | Source or reference | Identifiers | Additional information |
|---|---|---|---|---|
| Chemical compound, drug | EverbriteMounting Medium | Biotium | Cat#: 23,002 | Mounting medium with DAPI |
| Software, algorithm | FIJI | *Schindelin et al., 2012* DOI: 10.1038/nmeth.2019 | RRID:SCR_002285 | Image analysis software |
| Software, algorithm | GraphPad Pism 9.0 | GraphPad Pism 9.0 | RRID:SCR_002798 | Statistical analysis software |
| Other | 0.2 µm green fluorescent polystyrene beads | Degradex Phosphorex | Cat#: 2,108B | Fluorescent beads for bead feeding assays. |

