## [Editor Report]

Host factors involved in microsporidia infection have not been characterized in any detail, despite their ubiquitous distribution as infectious agents of animals. Through genetics screens, this study identifies a gene required for microsporidia invasion of the *C. elegans* intestine. Decreased susceptibility to microsporidia comes at the cost of increased colonization and infection by certain bacterial species, suggesting a tradeoff between resistance to different classes of pathogens.

---

## [Decision Letter]

**Decision letter after peer review:**

Thank you for submitting your article "An intestinally secreted host factor limits bacterial colonization but promotes microsporidia invasion of *C. elegans*" for consideration by *eLife*. Your article has been reviewed by 3 peer reviewers, and the evaluation has been overseen by a Reviewing Editor and Piali Sengupta as the Senior Editor. The reviewers have opted to remain anonymous.

Essential revisions:

Overall, the reviewers were highly enthusiasm about the discovery of a host factor necessary for infection with microsporidia, based on how little is known regarding these pathogens. The reviewers found the involvement of AAIM-1 in bacterial resistance less compelling, proposing the study remain centered on the role of the factor in relation to microsporidia. Considering that the molecular function of AAIM-1 remains poorly defined, the authors should also avoid language that too concretely defines the function of the factor. Essential revisions to the text are listed below, but no additional experimental data is required for publication.

(1) The authors should be more explicit about the modest effect of AAIM-1 disruption on resistance to bacterial infection. Discussion of tradeoffs between resistance to microsporidia and bacteria appear exaggerated based on the data presented. This includes modifying the title to reflect the weakness of the bacterial phenotype, e.g. "An intestinally secreted host factor promotes microsporidia invasion of *C. elegans*."

(2) Given the lack of information about the molecular function of AAIM-1, the authors should consider models that account for the stage-specific effects of the factor, rather than very direct effects on invasion.

*Combined Recommendations for the authors:*

(1) Images supporting the proposed mechanism by which aaim-1 modulates the orientation and hence invasive potential of microsporidia in the *C. elegans* intestine should be better defined.

(2) Discuss or document developmental effects that may be associated with infection with microsporidia. Figure 1a. Infection with microsporidia appears to be causing larval developmental arrest. The worms seem to hardly grow beyond L1 body length (250 micrometers). A careful analysis of body length in infected N2 animals would show whether they are arrested in L1 or L1 +L2 larva stage. The developmental arrest can also be ascertained using markers for germline and vulva development. Both larval arrest and smaller brood size are indicators of caloric restriction and starvation. The starvation in presence of microsporidia can be ascertained or ruled out by using markers of lipid breakdown such as acs-2, loss of lipid droplets, etc. It is likely that AAIM- bypasses the microsporidia-induced developmental arrest, observing the bottom row of images in Figure 1A suggests that this is the case since aaim-1 mutants don't show arrest of body length. Similar observations have been reported for E. faecalis infection of *C. elegans* [Garsin, Science 2003; Dasgupta, Micropublication 2020]. Authors could test if aaim-1 can alleviate infection mediated or caloric restriction mediated arrest.

(3) Is there published literature to show that spore orientation is causative or correlated with the firing of polar tubes and invasion?

---

## [Author Response]

Essential revisions:Overall, the reviewers were highly enthusiasm about the discovery of a host factor necessary for infection with microsporidia, based on how little is known regarding these pathogens. The reviewers found the involvement of AAIM-1 in bacterial resistance less compelling, proposing the study remain centered on the role of the factor in relation to microsporidia. Considering that the molecular function of AAIM-1 remains poorly defined, the authors should also avoid language that too concretely defines the function of the factor. Essential revisions to the text are listed below, but no additional experimental data is required for publication.

We thank the reviewers for their comments. We address the involvement of AAIM-1 in bacterial resistance below.

(1) The authors should be more explicit about the modest effect of AAIM-1 disruption on resistance to bacterial infection. Discussion of tradeoffs between resistance to microsporidia and bacteria appear exaggerated based on the data presented. This includes modifying the title to reflect the weakness of the bacterial phenotype, e.g. "An intestinally secreted host factor promotes microsporidia invasion of *C. elegans*."

We have changed the title to “An intestinally secreted host factor promotes microsporidia invasion of *C. elegans*“ to emphasize the role of AAIM-1 in microsporidia invasion and remove emphasis from the bacterial phenotype. We have also expanded upon our discussion in lines 324-333 describing how loss of AAIM-1 impacts the animal’s interactions with bacteria.

However, we feel that our previous draft provided several lines of evidence for the role of AAIM-1 in limiting *Pseudomonas aeruginosa* PA14 infection including (1) upregulation of *aaim-1* in the intestine in response to PA14 exposure (Figure 5a and S6f), (2) overexpression of AAIM-1 significantly decreasing PA14 burden in the intestine (Figure 5d and e), (3) loss of AAIM-1 leading to enhanced susceptibility and increased PA14 colonization (Figure 5b and c and S6a, b, and e), and (4) competition assays showing reduced reproductive fitness of *aaim-1* mutants on PA14 (Figure 6a).

To quantify the effect of loss of AAIM-1 on PA14 susceptibility, we have now repeated the survival curves of N2 and two *aaim-1* mutant alleles in comparison to the well-characterized p38 MAP kinase mutant *pmk-1(km25)* as a positive control for increased susceptibility to infection (Figure 5b and S6a and b). To compare the decrease in survival between WT and mutant animals, we have calculated the TD_50_ values of different strains (see figure legends of corresponding survival curves) and using these values have calculated the mutant strain’s relative survival (Figure S6c). This additional data demonstrates that *aaim-1* mutants survive ~78% as well as WT whereas the pmk-1 mutants survive ~51% as well.

In response to a comment by reviewer 2, we have now also assessed the role of *aaim-1* in defence against another bacterial pathogen, *Staphylococcus aureus*. We do not see a difference in survival between *aaim-1* alleles and N2, suggesting that AAIM-1 does not provide protection against all bacterial pathogens (Figure S7a).

(2) Given the lack of information about the molecular function of AAIM-1, the authors should consider models that account for the stage-specific effects of the factor, rather than very direct effects on invasion.

We have now modified the discussion in lines 319-321 to further discuss how AAIM-1 is functioning in a developmentally dependant fashion to promote microsporidia invasion.

Combined Recommendations for the authors:(1) Images supporting the proposed mechanism by which aaim-1 modulates the orientation and hence invasive potential of microsporidia in the *C. elegans* intestine should be better defined.

To better visualize the orientation of microsporidia spores in the intestinal lumen, we acquired a fluorescently tagged strain of *C. elegans* (GK288) carrying PGP-1::GFP (Sato et al. 2007). PGP-1 localizes to the apical membrane, providing a clear outline of the intestinal lumen. We crossed this strain into *aaim-1 (kea22)* and infected the animals with calcofluor white (CFW) stained spores. We have now replaced our previous images of invasion with ones showing the orientation of CFW stained spores relative to the PGP-1::GFP intestinal lumen at 63x magnification (Figure 4a).

(2) Discuss or document developmental effects that may be associated with infection with microsporidia. Figure 1a. Infection with microsporidia appears to be causing larval developmental arrest. The worms seem to hardly grow beyond L1 body length (250 micrometers). A careful analysis of body length in infected N2 animals would show whether they are arrested in L1 or L1 +L2 larva stage. The developmental arrest can also be ascertained using markers for germline and vulva development. Both larval arrest and smaller brood size are indicators of caloric restriction and starvation. The starvation in presence of microsporidia can be ascertained or ruled out by using markers of lipid breakdown such as acs-2, loss of lipid droplets, etc. It is likely that AAIM- bypasses the microsporidia-induced developmental arrest, observing the bottom row of images in Figure 1A suggests that this is the case since aaim-1 mutants don't show arrest of body length. Similar observations have been reported for E. faecalis infection of *C. elegans* [Garsin, Science 2003; Dasgupta, Micropublication 2020]. Authors could test if aaim-1 can alleviate infection mediated or caloric restriction mediated arrest.

The impact of *N. parisii* infection on *C. elegans* development has been previously documented both from the Troemel lab (Luallen et al. *PLOS One,* 2015 and Balla et al. *Nature Microbiology*, 2016) and our own lab (Willis et al., *Science Advances*, 2021 and Murareanu et al., *bioRxiv*, 2021). In particular, Supplementary Figure 1 of Willis et al., *Science Advances*, 2021 provides convincing evidence that the extent of developmental delay caused by *N. parisii* is directly correlated with the level of *N. parisii* infection. Thus, we think that *aaim-1* mutants are resistant to the effects of microsporidia infection due to their being significantly less infected. We have now added an additional reference in line 97 and a sentence to the results in lines 105-106 to describe this previously published result.

(3) Is there published literature to show that spore orientation is causative or correlated with the firing of polar tubes and invasion?

To date, almost all studies of how microsporidia invade cells has focused on in vitro cell models. Our in vivo characterization is the first time to our knowledge, that spore orientation and angle have been investigated. Therefore, our finding that both these properties are altered in the *aaim-1* mutant background, which possesses reduced invasion, suggests that these two properties are correlated.